# Validation of the Micronutrient and Environmental Enteric Dysfunction Assessment Tool and evaluation of biomarker risk factors for growth faltering and vaccine failure in young Malian children

**Michael B. Arndt**[1,2]\*, **Jason L. Cantera**[1], **Laina D. Mercer**[1], **Michael Kalnoky**[1], **Heather N. White**[1], **Gregory Bizilj**[1], **David S. Boyle**[1], **Eugenio L. de Hostos**[1], **Robert K. M. Choy**[1]

1 PATH, Seattle, Washington, United States of America, 2 Department of Global Health, University of Washington, United States of America

\* MArndt@uw.edu

## Abstract

Environmental enteric dysfunction (EED) is an intestinal disorder common among children in low-resource settings and is associated with increased risk of growth stunting, cognitive deficits, and reduced oral vaccine immunogenicity. The Micronutrient and EED Assessment Tool (MEEDAT) is a multiplexed immunoassay that measures biomarkers previously associated with child growth faltering and/or oral vaccine immunogenicity: intestinal fatty acid–binding protein (I-FABP), soluble CD14 (sCD14), insulin-like growth factor 1 (IGF-1), and fibroblast growth factor 21 (FGF21). MEEDAT also measures systemic inflammation (α1-acid glycoprotein, C-reactive protein), ferritin, soluble transferrin receptor, retinol binding protein 4, thyroglobulin, and *Plasmodium falciparum* antigenemia (histidine-rich protein 2). The performance of MEEDAT was compared with commercially available enzyme-linked immunosorbent assays (ELISAs) using 300 specimens from Malian infant clinical trial participants. Regression methods were used to test if MEEDAT biomarkers were associated with seroconversion to meningococcal A conjugate vaccine (MenAV), yellow fever vaccine (YFV), and pentavalent rotavirus vaccine (PRV) after 28 days, or with growth faltering over 12 weeks. The Pearson correlations between the MEEDAT and ELISA results were 0.97, 0.86, 0.80, and 0.97 for serum I-FABP, sCD14, IGF-1, and FGF21, respectively. There were significant associations between I-FABP concentration and the probability of PRV IgG seroconversion and between IGF-1 concentration and the probability of YFV seroconversion. In multivariable models neither association remained significant, however there was a significant negative association between AGP concentration and YFV seroconversion. GLP-2 and sCD14 concentrations were significantly negatively associated with 12-week change in weight-for-age z-score and weight-for-height z-score in multivariable models. MEEDAT performed well in comparison to commercially-available ELISAs for the measurement of four analytes for EED and growth hormone resistance. Adoption of MEEDAT in low-

**Data Availability Statement:** All relevant data are within the manuscript and its Supporting Information files.

**Funding:** RC, EdH, MA, JC, DS received funding from the government of the United Kingdom (https://www.gov.uk/government/organisations/department-for-international-development). The parent trial, RVI-PRV-01, was supported by the Bill and Melinda Gates Foundation (OPP1017334) https://www.gatesfoundation.org/. The funders had no role in study design, data collection and analysis, decision to publish, or preparation of the manuscript.

**Competing interests:** The authors have declared that no competing interests exist.

resource settings could help accelerate the identification of interventions that prevent or treat child stunting and interventions that boost the immunogenicity of child vaccinations.

## Author summary

Environmental enteric dysfunction (EED) is an intestinal disorder common among children in low-resource settings and has been associated with increased risk of growth stunting, cognitive deficits, and reduced oral vaccine immunogenicity. A key challenge to identifying children with EED at highest risk of morbid sequelae is the lack of validated predictive biomarkers. Ongoing clinical studies are testing and validating EED biomarkers in child populations at risk for stunting, yet testing multiple biomarkers commonly requires specialized equipment, complex methods, resources, and considerable effort. The Micronutrient and EED Assessment Tool (MEEDAT) is a multiplexed immunoassay that measures biomarkers associated with child growth faltering and oral vaccine immunogenicity, and biomarkers indicative of systemic inflammation and micronutrient deficiencies. The performance of MEEDAT was well-correlated with commercial monoplex assays in specimens from children living in a low-resource setting in the present study. MEEDAT biomarkers were associated with growth outcomes and seroconversion in response to several vaccines. MEEDAT has the potential to reduce the time and cost of evaluating impact of interventions targeting EED.

## Introduction

Environmental enteric dysfunction (EED) is an intestinal disorder common among people living in low-resource settings with a high enteric-pathogen burden and poor sanitation and hygiene [1, 2]. EED is characterized by mucosal inflammation, reduced barrier integrity, and malabsorption and may be associated with increased risk of growth stunting, reduced oral vaccine responsiveness, and impaired cognitive development in children in low- and middle-income countries [3–5]. Growth stunting (height-for-age z-score [HAZ] < –2) is associated with increased risk of chronic health issues including, but not limited to, cognitive deficits and increased susceptibility to infections [6–9]. EED thus directly and indirectly contributes to child mortality and morbidity due to other common childhood diseases such as pneumonia, acute diarrhea, and malaria. An estimated 22.2% of all children are stunted globally (approximately 150.8 million), and much of this burden is thought to be due to EED [10].

 A key challenge to identifying children with EED at highest risk of morbid sequelae is the lack of validated predictive biomarkers. We have developed a multiplexed planar immunoassay that measures four blood-based markers that are indicative of EED and growth-hormone resistance (in addition to seven markers of systemic inflammation and micronutrient status, described below). These biomarkers include intestinal fatty acid–binding protein (I-FABP), soluble CD14 (sCD14), insulin-like growth factor 1 (IGF-1), and fibroblast growth factor 21 (FGF21). I-FABP is a marker of small intestine epithelial damage that has been associated with stunting and future risk of growth faltering [11]. Elevated blood levels of sCD14 are a marker of monocyte activation (triggered by exposure to lipopolysaccharide) indicative of Gram-negative bacterial translocation across the intestinal epithelial barrier. It has been associated with future risk of growth faltering, poor immune responses to oral immunizations (oral polio vaccine type 1 and Rotarix rotavirus vaccine) in infants, and poor cognitive test scores later in

childhood [5, 11, 12]. IGF-1 is a hormone released by the liver in response to growth hormone (GH), whose binding to GH promotes tissue and bone growth. GH resistance, defined by elevated GH levels and conversely low levels of IGF-1, is exhibited in states of undernutrition, including chronic caloric insufficiency, protein deficiency, and isolated micronutrient deficiencies (zinc, vitamin A, and magnesium) [13–15]. GH resistance may mediate the relationship between these nutritional deficiencies and poor linear and ponderal growth in children. FGF21 is an endocrine hormone suggestive of GH resistance attributable to reduced protein intake [16, 17]. It has been associated with risk of growth faltering during nutritional supplementation [18]. A summary of this information and the clinical range from previous studies is provided (**S1 Table**). Although prior studies have observed relationships between immune response to orally administered vaccinations with EED and systemic inflammation biomarkers, few studies to date have tested whether biomarkers of GH resistance, systemic inflammation, or EED are associated with responses to parenteral vaccines [12]. No studies have tested if such biomarkers are associated with risk of natural rotavirus infection. A priori, we expect that EED would not be associated with parenteral vaccine response due to the lack of mucosal immune system involvement.

Quansys Biosciences (Logan, Utah, USA) specializes in the development and manufacture of Q-Plex arrays, or multiplex immunoassay technologies for biomarker detection and quantification. Q-Plex arrays are designed as a relatively low-cost method for the concurrent quantification of multiple markers using geometric planar arrays of target-specific antibodies in each well of a 96-well plate. The Q-Plex platform can quantify up to 18 analytes per well while requiring a relatively small volume of biological sample (e.g., 10 μl). We developed the 11-plex Micronutrient and EED Assessment Tool (MEEDAT, **Fig 1**) by adding I-FABP, sCD14, IGF-1, and FGF21 to the Q-Plex Human Micronutrient Array (7-plex) that measures micronutrient biomarkers (ferritin, soluble transferrin receptor [sTfR], retinol binding protein 4 [RBP4], thyroglobulin), acute and chronic inflammatory biomarkers (C-reactive protein [CRP] and α1-acid glycoprotein [AGP]), and histidine-rich protein 2 (HRP2), an indicator of *P. falciparum* antigenemia [19, 20]. Blood levels of serum ferritin and RBP4 are influenced by acute-phase inflammation and may be considered indirect biomarkers of inflammation [21, 22]. MEEDAT is intended to offer a rapid and effective tool for efficiently screening children for EED prior to enrollment into clinical trials of candidate EED interventions and will streamline the evaluation of efficacy in trials for which the biomarkers are clinical endpoints. MEEDAT has the potential to reduce the time and cost and increase the accuracy of blood-based biomarker measurement among children at risk for/suffering from malnutrition in clinical research and accelerate the identification and deployment of effective interventions. In order to justify its use in research studies, the performance of MEEDAT needed to be verified with well-qualified clinical specimens collected from children in a low-resource setting, a population at high risk for EED and stunting.

The multiplexing of protein-based biomarkers on the Q-Plex platform has several potential advantages over monoplex ELISAs or other multiplex platforms, including reductions in assay time, small sample volume requirement, complexity (versus bead-based approaches), cost, and ease of use. The Quansys Q-View Imager LS is well suited for use in low-resource settings due to its small footprint (20.3 x 25.4 x 55.9 cm), simplicity, and comparatively low cost. The generation of quantitative data on 11 biomarkers using the MEEDAT takes the same time and labor as a monoplex ELISA testing a single biomarker. Prior work has demonstrated the laboratory performance of the existing 7-plex Q-Plex Human Micronutrient Array versus monoplex ELISAs using samples from pregnant women in Niger and adult volunteers in the United States [19, 20, 23]. We conducted longitudinal, cross-sectional, and nested case-control studies using data and sera samples from 300 children enrolled in RVI-PRV-01, a phase IV, double-blind,

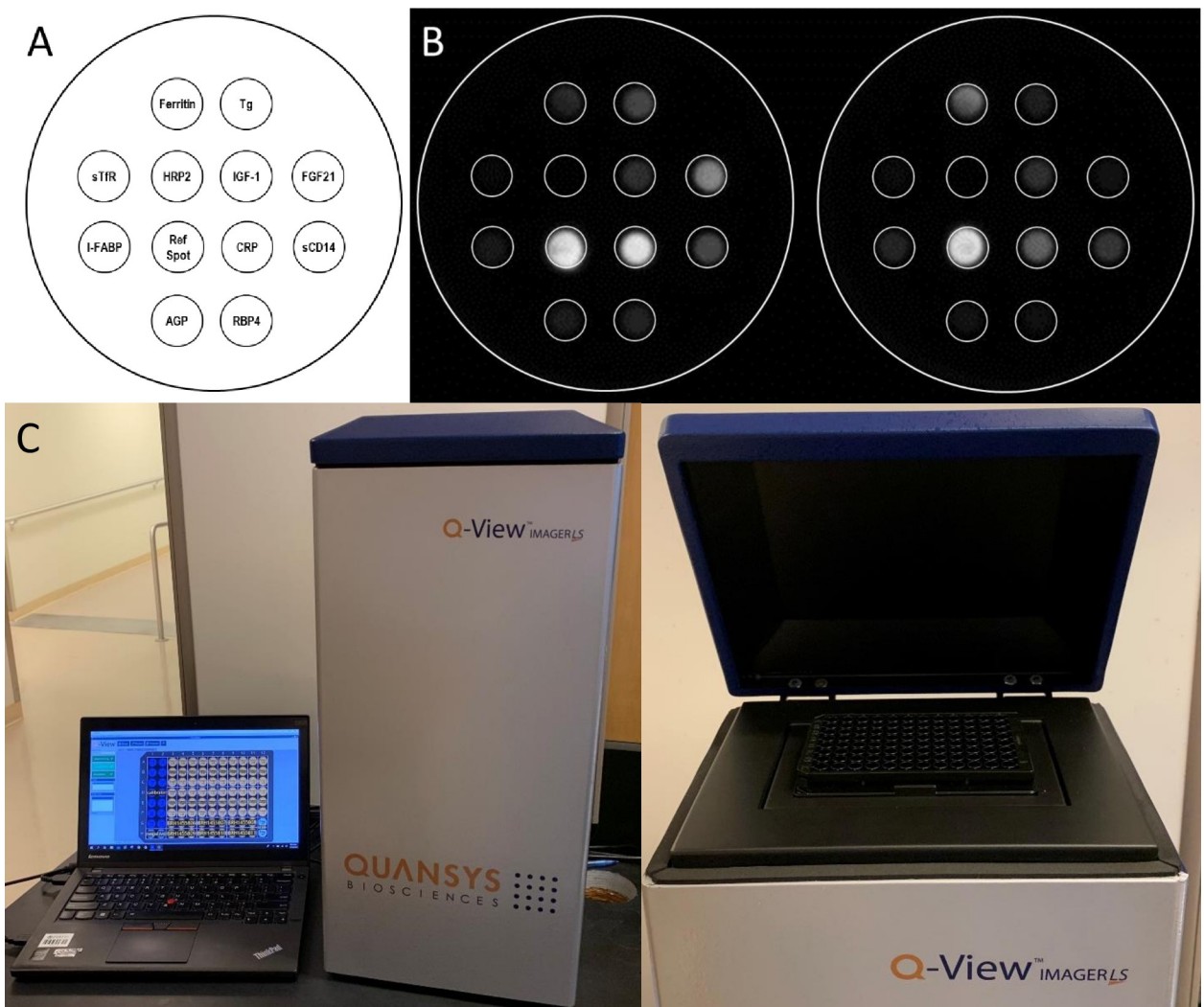

**Fig 1. The 11-plex Micronutrient and EED Assessment Tool.** (A) Schematic of placement of assays in each test well. (B) Images of developed signal in wells of a MEEDAT plate, with varying concentration of each analyte. (C) Images of Q-View Imager LS in use.

randomized controlled vaccine study in Mali. The prevalence of stunting in Mali was 30.4% in 2015 [24], and children living in this population are likely experiencing EED, but little EED data is available from this setting. The testing of specimens from children in this population helps to address the paucity of EED data and demonstrate the validity of the MEEDAT in comparison with monoplex assays. The primary study objective was to test the laboratory performance of the novel multiplex assay by using it to quantitate plasma concentrations of IGF-1, FGF21, sCD14, I-FABP in child specimens and then compare with concentrations measured by monoplex ELISAs. The secondary objectives were to test if biomarkers of EED, systemic inflammation, and growth hormone resistance at baseline were associated with seroconversion to oral rotavirus vaccination and 12-week height/weight growth. Finally, the exploratory objectives were to test if these biomarkers were associated with 1) seroconversion to parenteral meningococcus A or yellow fever vaccines and 2) natural rotavirus infection in children not given PRV.

## Methods

### Ethics statement

The original RVI-PRV-01 trial protocol was approved by the University of Maryland School of Medicine Institutional Review Board; the Ethical Committee of Faculté de Médecine, Pharmacie, et Odontostomatologie of Mali (FMPOS-EC); the Ministry of Health of Mali; Western Institutional Review Board (Puyallup, WA, USA); and leaders of the involved communities. Parents or guardians of participants provided written informed consent prior to initiation of study procedures. The trial was registered with ClinicalTrials.gov (registration number NCT02286895). The study protocol for the current nested study was reviewed and approved by FMPOS-EC and by PATH's Research Determination Committee.

### Study population

The study used data and serum specimens from a sub-sample (described below) of 300 children enrolled in RVI-PRV-01, an open-label phase IV, double-blind, randomized controlled study in Mali. RVI-PRV-01 evaluated the noninferiority of the immune responses to measles vaccine (MV), meningococcal conjugate vaccine (MenAV; PsA-TT-5μg), and yellow fever vaccine (YFV) given with oral pentavalent rotavirus vaccine (PRV; RotaTeq) compared to that given without PRV (non-PRV) in 600 healthy infants 9–11 months of age at enrollment. This study also evaluated the superiority of the immune response to a supplemental dose of PRV given at 9–11 months of age with MV, MenAV and YFV compared with no supplemental dose (non-PRV group). From 15 October 2014 to 18 December 2014, eligible infants aged 9–11 months were enrolled at nine health centers in Bamako, Mali. Eligible infants resided in the study area, were generally healthy, and had been fully vaccinated according to the local immunization schedule. Details on the exclusion criteria have been published previously [25].

Three hundred children from RVI-PRV-01 trial were selected for this study, including 246 randomly selected from the PRV group (Fig 2). The remaining 54 were selected from non-PRV children who did not have evidence of seroprotection from rotavirus at enrollment (i.e., baseline anti-rotavirus IgA and anti-rotavirus IgG titer < 20 units/ml). These 54 children were used for a rotavirus case-control comparison of baseline biomarker levels. All 27 children who exhibited a threefold increase in anti-rotavirus IgA or anti-rotavirus IgG between baseline and day 28 were selected as rotavirus cases. While laboratory confirmation of rotavirus infection was not performed, the rise in titer is considered a good proxy for natural infection in the absence of a dose of rotavirus vaccine. Twenty-seven children who did not exhibit a titer rise of this magnitude were randomly selected as uninfected controls from the remaining children without evidence of seroprotection at enrollment.

Data from all 300 children were included in the primary analysis comparing levels of I-FABP, sCD14, IGF-1, and FGF21 quantified using MEEDAT versus ELISAs. Additionally, data from all 300 children were included in analyses regarding relationships between biomarkers of systemic inflammation (AGP and CRP), enteric dysfunction (sCD14 and I-FABP), and growth hormone axis with 12-week growth, immune response to MenAV, and immune response to YFV.

The subgroup of 220 children from the PRV group who did not experience gastrointestinal illness (e.g., gastroenteritis, vomiting, or diarrhea) during the 28 days of follow-up were included in analysis of the relationships between the above biomarkers and IgA or IgG seroconversion (threefold increase in titer from baseline) 28 days after receipt of oral rotavirus vaccine. As the study took place during rotavirus season, children (n = 26) who both experienced gastrointestinal illness and seroconverted were excluded to reduce the likelihood of

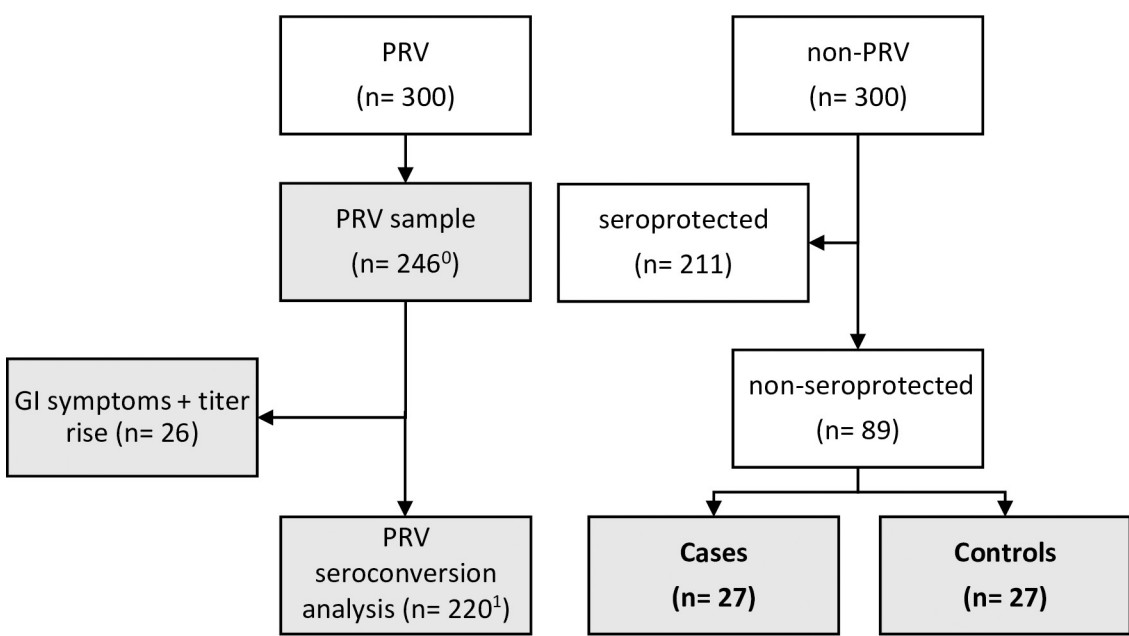

**Fig 2. Study design flow chart: Grey shading denotes specimens tested with MEEDAT and ELISAs, and inclusion in tests of biomarker associations with growth and parenteral vaccine seroconversion (YFV, MenAV). Bold text denotes inclusion in Case-Control study.** *Abbreviations*: GI, gastrointestinal; PRV, pentavalent rotavirus vaccine. [0] includes n = 152 tested with the GLP-2 ELISA. [1] includes n = 133 tested with the GLP-2 ELISA who did not experience GI symptoms and IgG or IgA titer rise.

misclassifying heightened rotavirus IgA or IgG at day 28 due to natural rotavirus infection as opposed to a response to vaccine.

## Data collection

In the RVI-PRV-01 trial, the parents were interviewed to collect baseline demographic (age and sex) and health information (known medical conditions and allergies), in order to confirm eligibility [25]. A study physician performed a physical examination of each infant prior to enrollment (day 0) and on days 7, 28, and 96 of follow-up. Physical exams included assessment of general appearance, height, weight, vital signs (body temperature, heart rate, respiratory rate), and physical examination of vital systems (i.e., chest auscultation, heart auscultation, abdomen palpation). Results were reviewed by study staff and with each subject's parents to confirm eligibility prior to conduct of further procedures. All children were tested for malaria infection by blood smear or rapid diagnostic test (RDT), using standard clinical RDTs in Mali; any infant with evidence of infection was provided treatment per local guidelines. Whenever an enrolled infant presented to a health center, the local hospital, or the Center for Vaccine Development Mali Study Center (CVD-Mali) during the 96-day follow-up period, a clinical assessment was performed by a physician per routine community practice. Data on subject illness history, physical examination, diagnosis, and treatment were collected.

All participants received MV by subcutaneous injection, YFV by intramuscular injection, and MenAV by intramuscular injection at a separate site [25]. Adverse events (AEs) such as fever, vomiting, diarrhea, and signs of potential intussusception were assessed through home visits and study clinic visits on days 1–5, 7, 28, 56, and 84, and unsolicited AEs were assessed until day 28. Serious adverse events were assessed from day 0 until day 84 of the study.

## Laboratory methods

**Specimen collection and preparation.**   Blood samples (3–5 mL) for serum fractionation were collected in serum separator tubes on days 0 and 28 for evaluation of serologic response to the administered vaccine antigens. Immediately after collection, the collection tube was inverted five times, labeled, and held upright for 30–120 minutes at room temperature to clot. All specimens were transported to the CVD-Mali laboratory for serum fractionation, which involved centrifugation of the blood collection tubes at 3,000 revolutions per minute for 5 minutes before collecting six serum aliquots of 200 μL. Remnant serum aliquots (~200–1,000 μL) were also collected, and all aliquots were stored at –20˚C until used. The day 0 specimens were subsequently shipped to Seattle, WA, USA, according to internationally recognized standards for transport of biospecimens.

**Quantitative measurement of EED biomarkers.**   Paired samples derived from the day 0 serum samples were tested using both the MEEDAT (Quansys Biosciences, Logan, UT, USA) and monoplex ELISAs for the four EED biomarkers described below. Serum specimens from the RVI-PRV-01 trial (N = 300) were tested for EED (I-FABP and sCD14) and GH resistance biomarkers (IGF-1 and FGF21) using both the MEEDAT and corresponding individual ELISAs to the four EED biomarkers as the predicate reference methods (described below). Initial MEEDAT testing performed on 80 specimens produced systematically invalid results for IGF-1, and therefore these specimens were re-run on MEEDAT plates from a second manufacturing lot (no. 19031914) in order to generate the IGF-1 data. Unfortunately, one of these specimens had insufficient volume for a second MEEDAT run, having been consumed in the monoplex ELISA testing. The remaining 220 specimens were run on MEEDAT plates from lot no. 19031914. There were insufficient specimen volumes to run sCD14 ELISAs for two participants.

A subset of 152 specimens were also tested using an ELISA for glucagon-like peptide 2 (GLP-2). GLP-2 is a biomarker that is under consideration for future assay development efforts as it is associated with stunting intestinal tissue repair and regeneration [26, 27].

**Measurement of immune response to vaccine antigens.**   In the RVI-PRV-01 trial, immune measures relevant to the administered vaccines were assessed in day 0 and day 28 specimens using standard measures, as described previously [25]. Yellow fever virus–specific neutralizing antibody titer (NT) was determined at the Robert Koch Institute (Berlin, Germany) using the Institute's methods [28]. A validated serum bactericidal assay (SBA) that uses infant rabbit complement was used to measure the titer of functional antibody in human sera to *Neisseria meningitidis* group A in the Vaccine Evaluation Unit laboratory at Public Health England (Manchester, UK). ELISAs were used to measure anti-rotavirus IgA and IgG levels in the Laboratory of Specialized Clinical Studies at the Cincinnati Children's Hospital Medical Center (Cincinnati, OH, USA). The positive control was pooled sera from subjects who had received a rotavirus vaccine or had experienced a natural rotavirus infection. The negative control was sera shown to have no antibody to rotavirus. Seroconversion for YFV (NT) and MenAV (SBA) was defined as fourfold or greater increase in titer from baseline. In the examination of rotavirus vaccine seroconversion, children who exhibited a threefold or greater increase in IgA or IgG titer from baseline were considered to have seroconverted; however, such children were excluded from the analysis of biomarker influence on rotavirus seroconversion if he/she experienced any gastrointestinal illness during the 28 days of follow-up, to reduce the likelihood of misclassifying heightened rotavirus IgA or IgG at day 28 due to natural rotavirus infection as response to vaccine.

## Biomarker testing

For the MEEDAT assay, the competitor mix was first added into the sample diluent. All samples and calibrator dilutions were diluted in sample diluent containing reconstituted

competitor as suggested in the manufacturer's protocol. An eight-point calibration curve (seven points plus one blank) was prepared by reconstituting the lyophilized calibrator using the complete sample diluent and then a further series of seven threefold dilutions. A 15 μL volume of each plasma sample was diluted in sample diluent to a final dilution of 1:10 for each sample. A 50 μL volume of calibrators and samples were added in duplicate or triplicate wells of the plate, respectively. From this point, the protocol proceeded as described in the manufacturer's kit instruction booklet. After addition of the standards and samples, each plate was incubated at room temperature for 2 hours with shaking using a Barnstead 4625 titer plate shaker (Thermo Scientific, Dubuque, Iowa, **USA**) at 500 revolutions per minute. All reactions were aspirated and washed three times with 350 μL of 1x Tris buffered saline—Tween 20 using an automatic plate washer (405 TS Microplate Washer, BioTek Instruments Inc., Winooski, Vermont, USA). After washing, 50 μL of detection mix was added to each well, and the plate was incubated with shaking for 1 hour and then washed as described above. Labeling was performed by adding 50 μL streptavidin-horseradish peroxidase solution to each well and shaken for 20 minutes. The solution was then aspirated, and the wells washed six times. The chemiluminescent substrates A and B were mixed in equal volumes, and 50 μL of the mixture was added to each well.

Each plate was then imaged at 270 seconds of exposure time using a Q-View Imager LS (Quansys Biosciences). Plate map overlay onto the analyte spots in each well, as well as measurement of the chemiluminescent signal from each spot in units of pixel intensity, was done using Q-View Software (Quansys Biosciences). The software then applied the pixel intensities for each spot in the wells into the calibrator concentration values and fitted using a five-parameter logistic curve setting. The pixel intensities of the spots in each test well were used to interpolate the concentration of each analyte relative to its calibrator curve. All the curve fitting and data reduction steps are automatically applied via the software. The assay ranges for each kit lot were applied to exclude values beyond the concentration ranges to yield precise concentration estimates. All values were adjusted for dilution used. Each analyte within the validated MEEDAT array met the intra-assay and inter-assay specifications of <10% and <15% coefficient of variation (CV), respectively.

Any samples that fell more than 20% below the lower limit of quantification (LLOQ) were assigned a value of one-half the LLOQ. This rule was also applied to output from standard ELISAs (described below). Those values that fell more than 20% above the upper limit of quantification were kept as those values but were not included in correlation analysis comparing the two quantification methods. No replacement values were included in such comparisons.

Each paired day 0 sample was also quantified in parallel using standard ELISAs for FGF21, IGF-1, sCD14 (all R&D Systems; Minneapolis, MN, USA), and I-FABP (Hycult Biotech; Wayne, PA, USA). Each ELISA was performed as per manufacturers' protocols, with each sample tested in duplicate. GLP-2 levels were quantified with an ELISA (MilliporeSigma; Darmstadt, Germany) per the manufacturer's recommended protocols in duplicate. The standard ELISA kits exhibited intra- and inter-assay precisions of <15% and <10% CV, respectively as calculated from each assay run. Serum GLP-2 measurement was done only in a subset of 152 children who were anti-rotavirus IgA seronegative (< 20 units/mL) at baseline in order to investigate associations between GLP-2 and rotavirus vaccine seroconversion as well as height and weight growth.

## Statistical methods

All statistical analyses were performed using Stata version 14.2 SE (College Station, TX, USA). World Health Organization (WHO) Anthro software was used to calculate z-scores (height-

for-age [HAZ], weight-for-age [WAZ], and weight-for-height [WHZ]) from raw anthropometric data using standards established in the 2006 WHO Multicentre Growth Reference Study Group [29]. Height observations were dropped from the data set if deemed implausible (i.e., $\geq 1$ cm lost from the baseline measurement). Just one observation was dropped from the growth analyses due to this implausibility criterion. No clear height gain outliers were observed with which to inform a similar rationale for removal.

For the initial laboratory-based performance verification and validation testing of MEE-DAT, we used scatter plots and Pearson correlations to evaluate whether IGF-1, FGF21, sCD14, and I-FABP results from the MEEDAT and ELISAs covaried linearly. Absolute levels were compared by expressing results from the 11-plex assay as a percentage of the results from conventional assays, and concentration-dependent bias in differences between measures from the MEEDAT and ELISAs were evaluated using Bland-Altman plots with the average concentration value for the two methods (x-axes) plotted against the absolute difference on the y-axis:

$$Y = \frac{\text{MEEDAT result} - \text{ELISA result}}{\text{average concentration}}$$

Biomarker concentrations were categorized into four quartiles based on the distribution of all measurements. The nonparametric test for trend, an extension of the Wilcoxon rank-sum test, was used to test for a dose-response relationship between the sera biomarkers and seroconversion. Relative risk (RR) regression using a Poisson working model with robust standard errors was further used to evaluate whether the seroconversions of the different vaccines are associated with biomarkers of EED, systemic inflammation, or growth hormone resistance [30]. All biomarkers were transformed to the log base 2 scale resulting in coefficients that could be interpreted as the effect of doubling the biomarker concentration. Analyses were run univariably and then with adjustment for other biomarkers and baseline titer.

Linear regression was also used to test the association between continuous log-2 transformed biomarker levels and subsequent linear and weight growth through 12 weeks of follow-up. Univariable analyses were run for each biomarker, and then analyses with adjustment for baseline and other pertinent MEEDAT biomarkers, including ferritin and retinol binding protein 4.

Among children who did not receive PRV at enrollment, with baseline anti-rotavirus IgG and IgA titers < 20 units/mL, Student's t-tests with unequal variances were used to compare the mean baseline biomarker concentrations between those who did and did not experience natural rotavirus infection during the 28 days following enrollment.

## Results

### Demographics, growth status, and immunization response

Samples from a total of 300 children were included from the parent RVI-PRV-01 study. Age at enrollment ranged from 273 to 364 days and 48% of subjects were female (**Table 1**). At enrollment, 18% of children were stunted and 1.7% were wasted (WHZ < −2). Median LAZ and WAZ were −1.06 and −0.79, respectively. Implausible height and weight data were omitted from one infant at the 12-week time point. Median LAZ declined to −1.26 after 12 weeks of follow-up, whereas median WAZ increased slightly to −0.64. After 12 weeks of follow-up, 21% were stunted and 3% were wasted. Twenty-eight days after vaccination, 204 (68%) and 284 (95%) of the 300 children had seroconverted for YFV and MenAV, respectively. Out of the 220 children who had received PRV at baseline, 110 (50%) and 124 (56%) had seroconverted for rotavirus based upon IgA or IgG titers, respectively, 28 days after vaccination. The 220 children in the PRV arm excludes the 26 seroconverters who experienced gastrointestinal illness in the period (plausible natural rotavirus infections).

**Table 1. Characteristics of Malian infant participants (N = 300 unless otherwise noted).**

| Category | Characteristic | n (%) or median (IQR) |
|---|---|---|
| | Female | 143 (47.7) |
| | Age at enrollment (days) | 282 (277, 299.5) |
| Anthropometry at enrollment | Length-for-age z-score | −1.06 (−1.62, −0.37) |
| | Stunted | 54 (18.0) |
| | Weight-for-age z-score | −0.79 (−1.36, −0.12) |
| | Weight-for-length z-score | −0.20 (−0.80, −0.46) |
| | Wasted | 5 (1.7) |
| Anthropometry at 12 weeks[1] | Length-for-age z-score | −1.26 (−1.93, −0.73) |
| | Stunted | 63 (21.1) |
| | Weight-for-age z-score | −0.64 (−1.33, 0.05) |
| | Weight-for-length z-score | 0.00 (−0.67, 0.76) |
| | Wasted | 9 (3.0) |
| **Serum biomarker** | **Measurement method** | **n (%) or median (IQR)** |
| I-FABP (pg/mL) | MEEDAT | 1052.1 (697.4, 1493.2) |
| | ELISA | 820.4 (504.2, 1200.3) |
| sCD14 (ng/mL) | MEEDAT | 1797.7 (1429.7, 2159.9) |
| | ELISA | 1615.7 (1297.2, 1867.2) |
| IGF-1 (ng/mL) | MEEDAT[2] | 18.2 (9.7, 30.6) |
| | ELISA | 19.7 (12.2, 27.8) |
| FGF21 (pg/mL) | MEEDAT | 164.6 (96.2, 345.8) |
| | ELISA | 163.6 (92.4, 345.7) |
| AGP (g/L) | MEEDAT | 0.85 (0.68, 1.09) |
| CRP (mg/L) | MEEDAT | 0.65 (0.30, 2.65) |
| Ferritin (µg/L) | MEEDAT | 8.89 (3.30, 24.43) |
| RBP4 (µmol/L) | MEEDAT | 1.56 (1.30, 1.87) |
| sTfR (mg/L) | MEEDAT | 20.26 (15.18, 31.12) |
| Tg (µg/L) | MEEDAT | 45.85 (29.85, 75.38) |
| HRP-2 (µg/L) | MEEDAT | 0.006 (0.006, 0.009) |
| GLP-2 (ng/mL) | ELISA[3] | 3.8 (3.0, 4.9) |

*Abbreviations*: AGP, α1-acid glycoprotein; CRP, C-reactive protein; FGF21, fibroblast growth factor 21; GLP-2, glucagon-like peptide 2; HRP-2, histidine rich protein 2; I-FABP, intestinal fatty acid–binding protein; IGF-1, insulin-like growth factor 1; IQR, interquartile range; RBP4, retinol binding protein 4; sCD14, soluble cluster of differentiation 14; sTfR, soluble transferrin receptor; Tg, thyroglobulin.

[1] n = 299 children with valid anthropometry from month three.

[2] n = 299 children with valid IGF-1 output from MEEDAT.

[3] n = 152 children were tested for GLP-2.

## MEEDAT performance testing

MEEDAT and the ELISA methods for the four EED and GH axis analytes produced similar concentration results, with the median concentrations and interquartile ranges displayed in **Table 1**. The I-FABP concentrations for 70 (23%) specimens and FGF21 concentrations for 18 (6%) specimens were below the respective ELISAs' range of quantification. The IGF-1 concentrations for 64 (22%) specimens were below the range of quantification for MEEDAT or ELISA; 24 were out of range for MEEDAT ($< 2.7$ ng/mL), 23 were out of range for ELISA ($< 7.5$ ng/mL), and 17 were out of range for both methods. The out-of-range values were set to one-half the lower limit of quantification (per the Methods section) for all non-correlation analyses where they were featured. There were strong correlations between the MEEDAT and

**Table 2. Pearson correlation coefficients (*r*) for EED and GH resistance biomarkers (MEEDAT vs. ELISA) in the 300 Mali-based sera specimens.**

|   | I-FABP[0] | sCD14[1] | IGF-1[2] | FGF21[3] |
|---|---|---|---|---|
| *r* | 0.972 | 0.859 | 0.797 | 0.972 |

*Abbreviations*: FGF21, fibroblast growth factor 21; I-FABP, intestinal fatty acid–binding protein; IGF-1, insulin-like growth factor 1; sCD14, soluble cluster of differentiation 14.

[0] n = 230 in range of the ELISA.

[1] n = 298 valid results for the sCD14 ELISA.

[2] n = 235 in range of the IGF-1 ELISA and MEEDAT.

[3] n = 282 in range of the ELISA and MEEDAT.

ELISA for results that were within quantification range for both assays (**Table 2**). The Pearson correlation coefficients were 0.972, 0.859, 0.797, and 0.972 for I-FABP, sCD14, IGF-1, and FGF21, respectively. The Bland-Altman plots suggest that the absolute difference between the two assays does not increase or decrease as the average value changes (**Fig 3**). The mean difference in concentrations measured by the two methods were 259.2 pg/ml, 231.1 ng/ml, 1.6 ng/ml, and 10.1 pg/ml for I-FABP, sCD14, IGF-1, and FGF21, respectively. More than 95 percent of the datapoints were within 2 SD of the mean difference for all 4 analytes. Several of the MEEDAT biomarkers were correlated with each other, most notably AGP with CRP and I-FABP with ferritin (**S2 Table**). In addition, AGP was negatively correlated with IGF-1 (as was CRP) and positively correlated with sCD14, I-FABP, and FGF21.

## MEEDAT biomarkers and rotavirus, YFV, and MenAV seroconversion

In univariable analyses of biomarker quartiles, I-FABP and sCD14 were negatively associated with the probability of rotavirus seroconversion based on day-28 anti-rotavirus IgG (trend test p = 0.006) and IgA (trend test p = 0.017), respectively (**Table 3**). Thirty-nine percent of children seroconverted based on IgG in the highest quartile of I-FABP levels, compared with 67% in the lowest quartile. Forty-one percent of children seroconverted based on IgA in the highest quartile of sCD14 levels, compared with 57% in the lowest quartile. I-FABP and sCD14 also showed some negative trends for association for IgA and IgG seroconversion, respectively, but these trend tests were not statistically significant (p = 0.078 and 0.055, respectively). There was no evidence of associations for probability of rotavirus seroconversion with IGF-1, FGF21, AGP, CRP, or GLP-2. IGF-1 was positively associated with the probability of YFV seroconversion based on NT (trend test p < 0.001) in univariable quartile analysis (**Table 4**). Eighty-one percent of children seroconverted based on IgG in the highest quartile of IGF-1 levels, compared with 50% in the lowest quartile. I-FABP also showed a trend for a positive association with probability of YFV seroconversion, but the trend test was not statistically significant (trend test p = 0.060). There was no evidence of associations for probability of YFV seroconversion with sCD14, FGF21, AGP, CRP, or GLP-2. There was no evidence of associations for probability of MenAV seroconversion based on SBA with any of the biomarkers tested. In repeated analyses using I-FABP, sCD14, IGF-1, and FGF21 data from the ELISAs (**S3 Table** and **S4 Table**) there were a few differences in the results. FGF21 was negatively associated with the probability of rotavirus seroconversion based on anti-rotavirus IgA (p = 0.035) and sCD14 was negatively associated with the probability of rotavirus seroconversion based on anti-rotavirus IgG (p = 0.036) but not anti-rotavirus IgA (p = 0.091).

In RR regression analyses of rotavirus seroconversion, log-2 transformed I-FABP was significantly negatively associated with the probability of rotavirus seroconversion based on anti-

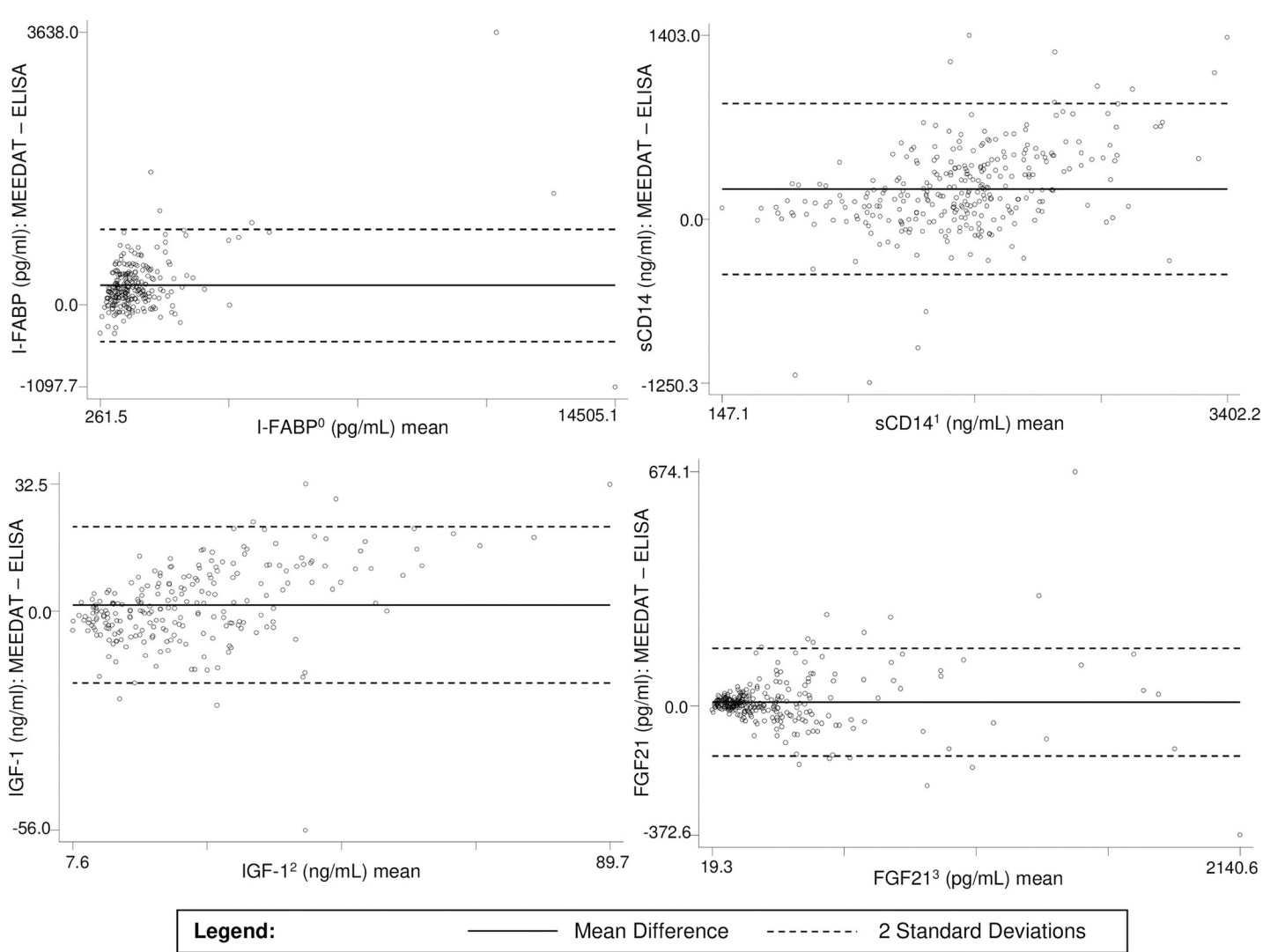

**Fig 3. Bland-Altman plots comparing MEEDAT to ELISA for I-FABP, sCD14, IGF-1, and FGF21 in 300 pediatric specimens from Mali.** *Abbreviations*: FGF21, fibroblast growth factor 21; I-FABP, intestinal fatty acid–binding protein; IGF-1, insulin-like growth factor 1; sCD14, soluble cluster of differentiation 14. [0] n = 230 in range of the I-FABP ELISA. [1] n = 298 valid results for the sCD14 ELISA. [2] n = 235 in range of the IGF-1 ELISA and MEEDAT. [3] n = 282 in range of the FGF21 ELISA and MEEDAT.

rotavirus IgG in the univariable analysis, but the association was not statistically significant in the multivariable analysis (adjusted RR = 0.88, 95% CI: 0.77, 1.00) (**Table 5**). There was no evidence of associations for probability of rotavirus seroconversion with log-2 transformed sCD14, IGF-1, FGF21, AGP, CRP, or GLP-2. Log-2 transformed IGF-1 was significantly positively associated with the probability of YFV seroconversion based on NT in the univariable RR regression analysis, but the association was not statistically significant in the multivariable analysis (adjusted RR = 1.04, 95% CI: 0.99, 1.10). Log-2 transformed AGP was significantly negatively associated with the probability of YFV seroconversion based on NT (adjusted RR = 0.87, 95% CI: 0.75, 1.00) in the multivariable RR regression analysis. For every doubling of AGP, there was a 13% reduction in the probability of YFV NT seroconversion. There was no evidence of associations for probability of YFV seroconversion with I-FABP, sCD14, FGF21, CRP, or GLP-2. There was no evidence of associations for probability of MenAV

**Table 3. Rotavirus vaccine seroconversion at 28 days post-immunization by EED, GH, and systemic inflammation status measured by MEEDAT, in infants without natural rotavirus infection during the 28 days of follow-up (n = 220 unless otherwise noted).**

| Biomarker and quartile | No. of infants (% of total) | No. (%) seroconverted (IgA ≥ threefold increase) | Trend test P-value | No. (%) seroconverted (IgG ≥ threefold increase) | Trend test P-value |
|---|---|---|---|---|---|
| **I-FABP—quartiles (cutoffs in pg/mL)** | | | | | |
| **1** ($<$ 697) | 51 (23.2) | 29 (56.9) | 0.078 | 34 (66.7) | **0.006** |
| **2** ($<$ 1052) | 50 (22.7) | 26 (52.0) | | 30 (60.0) | |
| **3** ($<$ 1493) | 62 (28.2) | 33 (53.2) | | 38 (61.3) | |
| **4** ($>$ 1493) | 57 (25.9) | 22 (38.6) | | 22 (38.6) | |
| **sCD14—quartiles (cutoffs in ng/mL)** | | | | | |
| **1** ($<$ 1429.7) | 51 (23.2) | 29 (56.9) | **0.017** | 34 (66.7) | 0.055 |
| **2** ($<$ 1797.7) | 55 (25.0) | 35 (63.6) | | 34 (61.8) | |
| **3** ($<$ 2159.9) | 53 (24.1) | 21 (39.6) | | 24 (45.3) | |
| **4** ($>$ 2159.9) | 61 (27.7) | 25 (41.0) | | 32 (52.5) | |
| **IGF-1—quartiles (cutoffs in ng/mL)[0]** | | | | | |
| **1** ($<$ 9.7) | 58 (26.4) | 31 (53.5) | 0.249 | 36 (62.1) | 0.615 |
| **2** ($<$ 18.2) | 51 (23.2) | 27 (52.9) | | 27 (52.9) | |
| **3** ($<$ 30.6) | 53 (24.1) | 28 (52.8) | | 27 (50.9) | |
| **4** ($>$ 30.6) | 57 (25.9) | 24 (42.1) | | 33 (57.9) | |
| **FGF21—quartiles (cutoffs in pg/mL)** | | | | | |
| **1** ($<$ 96.2) | 48 (21.8) | 28 (58.3) | 0.117 | 28 (58.3) | 0.344 |
| **2** ($<$ 164.6) | 49 (22.3) | 26 (53.1) | | 30 (61.2) | |
| **3** ($<$ 345.8) | 60 (27.3) | 28 (46.7) | | 34 (56.7) | |
| **4** ($>$ 345.8) | 63 (28.6) | 28 (44.4) | | 32 (50.8) | |
| **AGP—quartiles (cutoffs in g/L)** | | | | | |
| **1** ($<$ 0.68) | 52 (23.6) | 26 (50) | 0.246 | 30 (57.7) | 0.328 |
| **2** ($<$ 0.85) | 53 (24.1) | 32 (60.4) | | 34 (64.2) | |
| **3** ($<$ 1.09) | 61 (27.7) | 29 (47.5) | | 32 (52.5) | |
| **4** ($>$ 1.09) | 54 (24.5) | 23 (42.6) | | 28 (51.9) | |
| **CRP—quartiles (cutoffs in mg/L)** | | | | | |
| **1** ($<$ 0.30) | 54 (24.5) | 26 (48.2) | 0.259 | 30 (55.6) | 0.311 |
| **2** ($<$ 0.65) | 47 (21.4) | 32 (68.1) | | 34 (72.3) | |
| **3** ($<$ 2.39) | 59 (26.8) | 25 (42.4) | | 28 (47.5) | |
| **4** ($>$ 2.39) | 60 (27.3) | 27 (45.0) | | 32 (53.3) | |
| **GLP-2—quartiles (cutoffs in ng/mL)[1]** | | | | | |
| **1** ($<$ 3.04) | 33 (24.8) | 18 (54.7) | 0.787 | 24 (72.73) | 0.435 |
| **2** ($<$ 3.82) | 34 (25.6) | 25 (73.5) | | 23 (67.65) | |
| **3** ($<$ 4.85) | 33 (24.8) | 21 (63.6) | | 25 (75.76) | |
| **4** ($>$ 4.85) | 33 (24.8) | 18 (54.6) | | 20 (60.61) | |

*Abbreviations*: AGP, α1-acid glycoprotein; CI, confidence interval; CRP, C-reactive protein; FGF21, fibroblast growth factor 21; GLP-2, glucagon-like peptide 2; I-FABP, intestinal fatty acid–binding protein; IGF-1, insulin-like growth factor 1; IgA, immunoglobulin A; IgG, immunoglobulin G; sCD14, soluble cluster of differentiation 14.

[0] n = 299 for IGF-1.

[1] GLP-2 was tested in a subset of n = 152, only n = 133 of whom did not experience natural rotavirus infection during the 28 days of follow-up and are included in this table.

seroconversion based on SBA with any of the biomarkers tested. In repeated analyses using I-FABP, sCD14, IGF-1, and FGF21 data from the ELISAs (**S5 Table**) a few major differences were observed. In univariable analysis, both sCD14 and FGF21 levels were significantly negatively associated with the probability of rotavirus IgA seroconversion (p = 0.018 and p = 0.035,

**Table 4. Yellow fever and meningococcal A vaccine seroconversion at 28 days post-immunization by EED, GH, and systemic inflammation status measured by MEEDAT (N = 300 unless otherwise noted).**

| Biomarker and quartile | No. of infants (% of total) | No. (%) seroconverted to YFV (NT $\geq$ fourfold increase) | Trend test P-value | No. (%) seroconverted to MenAV (SBA $\geq$ fourfold increase) | Trend test P-value |
|---|---|---|---|---|---|
| **I-FABP—quartiles (cutoffs in pg/mL)** | | | | | |
| **1** ($<$ 697) | 75 (25.0) | 50 (66.7) | 0.060 | 71 (94.7) | 0.646 |
| **2** ($<$ 1052) | 75 (25.0) | 44 (58.7) | | 72 (96.0) | |
| **3** ($<$ 1493) | 75 (25.0) | 51 (68.0) | | 71 (94.7) | |
| **4** ($>$ 1493) | 75 (25.0) | 59 (78.7) | | 70 (93.3) | |
| **sCD14—quartiles (cutoffs in ng/mL)** | | | | | |
| **1** ($<$ 1429.7) | 75 (25.0) | 47 (62.7) | 0.122 | 72 (96.0) | 0.251 |
| **2** ($<$ 1797.7) | 75 (25.0) | 46 (61.3) | | 72 (96.0) | |
| **3** ($<$ 2159.9) | 75 (25.0) | 59 (78.7) | | 71 (94.7) | |
| **4** ($>$ 2159.9) | 75 (25.0) | 52 (69.3) | | 69 (92.0) | |
| **IGF-1—quartiles (cutoffs in ng/mL)[0]** | | | | | |
| **1** ($<$ 9.7) | 76 (25.4) | 38 (50.0) | $<$ **0.001** | 73 (96.1) | 0.795 |
| **2** ($<$ 18.2) | 74 (24.8) | 44 (59.5) | | 71 (96.0) | |
| **3** ($<$ 30.6) | 75 (25.1) | 61 (81.3) | | 67 (89.3) | |
| **4** ($>$ 30.6) | 74 (24.8) | 60 (81.1) | | 72 (97.3) | |
| **FGF21—quartiles (cutoffs in pg/mL)** | | | | | |
| **1** ($<$ 96.2) | 75 (25.0) | 50 (66.7) | 0.740 | 71 (94.7) | 0.359 |
| **2** ($<$ 164.6) | 75 (25.0) | 55 (73.3) | | 72 (96.0) | |
| **3** ($<$ 345.8) | 75 (25.0) | 49 (65.3) | | 73 (97.3) | |
| **4** ($>$ 345.8) | 75 (25.0) | 50 (66.7) | | 68 (90.7) | |
| **AGP—quartiles (cutoffs in g/L)** | | | | | |
| **1** ($<$ 0.68) | 75 (25.0) | 53 (70.7) | 0.486 | 74 (98.7) | 0.097 |
| **2** ($<$ 0.85) | 78 (26.0) | 53 (68.0) | | 74 (94.9) | |
| **3** ($<$ 1.09) | 73 (24.3) | 50 (68.5) | | 67 (91.8) | |
| **4** ($>$ 1.09) | 74 (24.7) | 48 (64.9) | | 69 (93.2) | |
| **CRP—quartiles (cutoffs in mg/L)** | | | | | |
| **1** ($<$ 0.30) | 79 (26.3) | 49 (62.0) | 0.255 | 76 (96.2) | 0.333 |
| **2** ($<$ 0.65) | 72 (24.0) | 51 (70.9) | | 67 (93.1) | |
| **3** ($<$ 2.39) | 74 (24.7) | 50 (67.6) | | 73 (98.7) | |
| **4** ($>$ 2.39) | 75 (25.0) | 54 (72.0) | | 68 (90.7) | |
| **GLP-2—quartiles (cutoffs in ng/mL)[1]** | | | | | |
| **1** ($<$ 3.04) | 38 (25.0) | 22 (57. 9) | 0.259 | 38 (100.0) | 0.388 |
| **2** ($<$ 3.82) | 38 (25.0) | 23 (60.5) | | 35 (92.1) | |
| **3** ($<$ 4.85) | 38 (25.0) | 26 (68.4) | | 36 (94.7) | |
| **4** ($>$ 4.85) | 38 (25.0) | 26 (68.4) | | 36 (94.7) | |

*Abbreviations*: AGP, α1-acid glycoprotein; CI, confidence interval; CRP, C-reactive protein; FGF21, fibroblast growth factor 21; GLP-2, glucagon-like peptide 2; I-FABP, intestinal fatty acid–binding protein; IGF-1, insulin-like growth factor 1; NT, neutralizing antibody titer; SBA, serum bactericidal assay; sCD14, soluble cluster of differentiation 14.

[0] n = 299 for IGF-1.

[1] n = 152 for GLP-2.

respectively), and I-FABP levels were not significantly associated with the probability of rotavirus IgG seroconversion (p = 0.072). In multivariable analyses, IGF-1 levels were significantly negatively associated with probability of rotavirus IgA seroconversion (p = 0.036) and AGP levels were not significantly associated with probability of YFV seroconversion (p = 0.075).

**Table 5. Associations between log-2 transformed concentrations of serum biomarkers at baseline with seroconversion to rotavirus, yellow fever, and meningococcal A vaccines 28 days post-dose.**

| | | Crude RR | (95% CI) | P-value | Adjusted RR[0] | (95% CI) | P-value |
|---|---|---|---|---|---|---|---|
| **Rotavirus seroconversion (IgA ≥ threefold increase)[1]** | I-FABP | 0.92 | (0.80, 1.06) | 0.265 | 1.01 | (0.83, 1.24) | 0.896 |
| | sCD14 | 0.91 | (0.76, 1.10) | 0.338 | 0.97 | (0.85, 1.10) | 0.610 |
| | IGF-1 | 0.96 | (0.89, 1.04) | 0.300 | 0.97 | (0.89, 1.04) | 0.387 |
| | FGF21 | 0.92 | (0.85, 1.00) | 0.059 | 0.94 | (0.86, 1.02) | 0.139 |
| | AGP | 0.96 | (0.78, 1.17) | 0.680 | 1.10 | (0.83, 1.46) | 0.507 |
| | CRP | 0.98 | (0.93, 1.03) | 0.400 | 0.98 | (0.93, 1.04) | 0.586 |
| | GLP-2[3] | 1.05 | (0.82, 1.36) | 0.691 | 1.07 | (0.82, 1.41) | 0.602 |
| **Rotavirus seroconversion (IgG ≥ threefold increase)[1]** | I-FABP | 0.88 | (0.78, 1.00) | **0.046** | 0.88 | (0.77, 1.00) | 0.057 |
| | sCD14 | 1.00 | (0.84, 1.19) | 0.975 | 1.17 | (0.93, 1.48) | 0.182 |
| | IGF-1 | 0.97 | (0.91, 1.04) | 0.421 | 0.98 | (0.92, 1.05) | 0.626 |
| | FGF21 | 0.96 | (0.89, 1.04) | 0.286 | 0.96 | (0.88, 1.05) | 0.428 |
| | AGP | 0.95 | (0.78, 1.15) | 0.596 | 1.01 | (0.77, 1.33) | 0.926 |
| | CRP | 0.98 | (0.93, 1.02) | 0.331 | 0.99 | (0.94, 1.04) | 0.627 |
| | GLP-2[3] | 0.97 | (0.79, 1.20) | 0.782 | 1.08 | (0.86, 1.37) | 0.499 |
| **Yellow fever seroconversion (NT ≥ fourfold increase)[2]** | I-FABP | 1.06 | (0.98, 1.15) | 0.123 | 1.04 | (0.97, 1.11) | 0.317 |
| | sCD14 | 1.06 | (0.93, 1.22) | 0.381 | 1.08 | (0.95, 1.22) | 0.259 |
| | IGF-1 | 1.11 | (1.04, 1.18) | **0.001** | 1.04 | (0.99, 1.10) | 0.110 |
| | FGF21 | 1.00 | (0.95, 1.05) | 0.936 | 0.99 | (0.95, 1.04) | 0.719 |
| | AGP | 0.94 | (0.86, 1.03) | 0.158 | 0.87 | (0.75, 1.00) | **0.049** |
| | CRP | 1.01 | (0.97, 1.04) | 0.752 | 1.02 | (0.98, 1.05) | 0.359 |
| | GLP-2[4] | 1.11 | (0.90, 1.36) | 0.337 | 1.14 | (0.92, 1.41) | 0.223 |
| **Meningococcus A seroconversion (SBA ≥ fourfold increase)[2]** | I-FABP | 0.98 | (0.94, 1.01) | 0.214 | 0.98 | (0.94, 1.03) | 0.439 |
| | sCD14 | 0.98 | (0.95, 1.01) | 0.278 | 1.00 | (0.96, 1.04) | 0.964 |
| | IGF-1 | 1.00 | (0.98, 1.01) | 0.556 | 1.00 | (0.99, 1.02) | 0.805 |
| | FGF21 | 0.99 | (0.97, 1.02) | 0.699 | 1.00 | (0.97, 1.03) | 0.974 |
| | AGP | 0.97 | (0.94, 1.00) | 0.066 | 0.98 | (0.94, 1.03) | 0.504 |
| | CRP | 1.00 | (0.98, 1.01) | 0.395 | 1.00 | (0.98, 1.01) | 0.645 |
| | GLP-2[4] | 0.97 | (0.92, 1.03) | 0.319 | 0.98 | (0.92, 1.04) | 0.467 |

*Abbreviations*: AGP, α1-acid glycoprotein; CI, confidence interval; CRP, C-reactive protein; FGF21, fibroblast growth factor 21; GLP-2, glucagon-like peptide 2; I-FABP, intestinal fatty acid–binding protein; IGF-1, insulin-like growth factor 1; IgA, immunoglobulin A; IgG, immunoglobulin G; NT, neutralizing antibody titer; RR, relative risk; SBA, serum bactericidal assay; sCD14, soluble cluster of differentiation 14.

[0] Estimates were adjusted for other log-transformed biomarkers (except GLP-2), age of infant (in months) at vaccination, relevant baseline log-transformed immunological titer(s).

[1] n = 220 children for rotavirus seroconversion analyses: excluding those with increased titer who reported diarrhea, gastroenteritis, or vomiting in the period; n = 219 for IGF-1 univariable and all adjusted RR (excluding GLP-2).

[2] N = 300; n = 299 for IGF-1 univariable and adjusted models (excluding GLP-2).

[3] n = 133 for GLP-2 univariable RR for rotavirus seroconversion; n = 132 for adjusted RR.

[4] n = 152 for GLP-2 univariable RR for YFV and MenAV seroconversion; n = 151 for adjusted RR.

## MEEDAT biomarkers and 12-week growth

In univariable analyses of the association between growth measurements (i.e., change in HAZ, WAZ, and WHZ) and biomarkers of EED, GH axis, systemic inflammation, and micronutrients, there was a significant positive association between log-2 transformed ferritin levels with 12-week change in WAZ (**Table 6**). The ferritin association did not maintain statistical significance in the multivariable model. There were significant negative associations between log-2

**Table 6. Associations between log-2 transformed concentrations of serum biomarkers at baseline with change in HAZ, WAZ, and WHZ over 12 weeks following enrollment.**

|  |  | Crude coefficient | (95% CI) | P-value | Adjusted coefficient[0] | (95% CI) | P-value |
|---|---|---|---|---|---|---|---|
| **ΔHAZ** | I-FABP | 0.04 | (−0.02, 0.10) | 0.230 | 0.02 | (−0.04, 0.09) | 0.480 |
|  | sCD14 | 0.06 | (−0.03, 0.15) | 0.205 | 0.04 | (−0.09, 0.16) | 0.557 |
|  | IGF-1[1] | −0.03 | (−0.07, 0.01) | 0.144 | −0.01 | (−0.05, 0.03) | 0.650 |
|  | FGF21 | 0.01 | (−0.03, 0.04) | 0.731 | −0.03 | (−0.06, 0.01) | 0.185 |
|  | AGP | 0.08 | (−0.01, 0.17) | 0.098 | 0.07 | (−0.05, 0.20) | 0.254 |
|  | CRP | 0.01 | (−0.02, 0.03) | 0.626 | −0.01 | (−0.04, 0.01) | 0.338 |
|  | GLP-2[2] | 0.11 | (−0.03, 0.24) | 0.132 | 0.11 | (−0.03, 0.24) | 0.112 |
|  | Ferritin | 0.01 | (−0.02, 0.03) | 0.706 | 0.01 | (−0.02, 0.03) | 0.706 |
|  | RBP4 | −0.01 | (−0.17, 0.14) | 0.860 | −0.01 | (−0.17, 0.14) | 0.860 |
| **ΔWAZ** | I-FABP | 0.02 | (−0.04, 0.08) | 0.456 | −0.01 | (−0.08, 0.06) | 0.771 |
|  | sCD14 | −0.04 | (−0.14, 0.06) | 0.455 | −0.15 | (−0.31, 0.00) | **0.046** |
|  | IGF-1[1] | −0.02 | (−0.06, 0.03) | 0.427 | 0.00 | (−0.05, 0.05) | 0.947 |
|  | FGF21 | 0.03 | (−0.02, 0.07) | 0.269 | 0.02 | (−0.03, 0.07) | 0.446 |
|  | AGP | 0.09 | (−0.02, 0.19) | 0.116 | 0.12 | (−0.05, 0.29) | 0.176 |
|  | CRP | 0.01 | (−0.01, 0.04) | 0.339 | 0.00 | (−0.04, 0.03) | 0.904 |
|  | GLP-2[2] | −0.15 | (−0.29, −0.01) | **0.037** | −0.22 | (−0.36, −0.08) | **0.002** |
|  | Ferritin | 0.04 | (0.00, 0.07) | **0.029** | 0.03 | (−0.01, 0.07) | 0.099 |
|  | RBP4 | 0.03 | (−0.09, 0.16) | 0.592 | 0.09 | (−0.10, 0.27) | 0.355 |
| **ΔWHZ** | I-FABP | 0.01 | (−0.08, 0.09) | 0.860 | −0.04 | (−0.14, 0.06) | 0.406 |
|  | sCD14 | −0.09 | (−0.22, 0.04) | 0.190 | −0.22 | (−0.43, −0.01) | **0.041** |
|  | IGF-1[1] | 0.00 | (−0.07, 0.07) | 0.998 | 0.02 | (−0.05, 0.09) | 0.568 |
|  | FGF21 | 0.03 | (−0.03, 0.10) | 0.335 | 0.04 | (−0.04, 0.11) | 0.320 |
|  | AGP | 0.07 | (−0.10, 0.23) | 0.435 | 0.16 | (−0.07, 0.39) | 0.169 |
|  | CRP | 0.01 | (−0.03, 0.05) | 0.532 | 0.00 | (−0.05, 0.05) | 0.961 |
|  | GLP-2[2] | −0.29 | (−0.50, −0.08) | **0.007** | −0.39 | (−0.60, −0.18) | **<0.001** |
|  | Ferritin | 0.04 | (−0.01, 0.09) | 0.095 | 0.03 | (−0.02, 0.07) | 0.280 |
|  | RBP4 | −0.01 | (−0.20, 0.18) | 0.884 | 0.09 | (−0.18, 0.36) | 0.516 |

*Abbreviations*: AGP, α1-acid glycoprotein; CI, confidence interval; CRP, C-reactive protein; FGF21, fibroblast growth factor 21; GLP-2, glucagon-like peptide 2; I-FABP, intestinal fatty acid–binding protein; IGF-1, insulin-like growth factor 1; RBP4, retinol binding protein-4; sCD14, soluble cluster of differentiation 14.

[0] Estimates were adjusted for other log-transformed biomarkers (except for GLP-2), sex, days of diarrhea in the first 28 days, relevant baseline anthropometric measurement (i.e., HAZ, WAZ, or WHZ).

[1] n = 299 children with valid anthropometric measurements at day 84 of follow-up; n = 298 for IGF-1 univariable and all adjusted models (excluding GLP-2).

[2] n = 151 who were tested for GLP-2 with valid anthropometric measurements at day 84 of follow-up are featured in crude and adjusted models for GLP-2.

transformed GLP-2 levels with 12-week change in WAZ and WHZ in univariable analyses restricted to the 151 children tested for GLP-2 with valid anthropometric measurements. The GLP-2 associations with WHZ and WAZ remained significant in multivariable models. For every doubling of GLP-2 level, there was an average of 0.22 SD decrease in WAZ (95% CI: −0.36, −0.08). For every doubling of GLP-2 level, there was an average of 0.39 SD decrease in WAZ (95% CI: −0.60, −0.18). In multivariable analyses, there were negative associations between log-2 transformed sCD14 levels with 12-week change in WAZ and WHZ. For every doubling of sCD14 level, there was an average of 0.15 SD decrease in WAZ (95% CI: −0.31, 0.00) and 0.22 SD decrease in WHZ (95% CI: −0.43, −0.01). There was no evidence of associations with 12-week change in HAZ, WAZ, or WHZ for any of the other biomarkers tested. All analyses were repeated using I-FABP, sCD14, IGF-1, and FGF21 data from the ELISAs (**S6**

**Table 7. Comparison of baseline serum biomarker concentrations between children with and without putative rotavirus infection within 28 days of enrollment.**

| | Infected[0] (n = 27) | | Uninfected[1] (n = 27) | | |
|---|---|---|---|---|---|
| | mean | (95% CI) | mean | (95% CI) | T-test[2] P-value |
| I-FABP (pg/mL) | 1667.6 | (627.8, 2707.5) | 1150.6 | (875.2, 1426.1) | 0.3312 |
| sCD14 (ng/mL) | 1651.12 | (1356.39, 1945.86) | 1704.36 | (1474.04, 1934.68) | 0.7711 |
| IGF-1 (ng/mL) | 19.31 | (13.73, 24.88) | 22.79 | (16.71, 28.86) | 0.5136 |
| FGF21 (pg/mL) | 186.9 | (111.9, 261.8) | 232.1 | (112.7, 351.5) | 0.3895 |
| AGP (g/L) | 0.92 | (0.78, 1.06) | 0.82 | (0.71, 0.92) | 0.2155 |
| CRP (mg/L) | 7.13 | (0.45, 13.80) | 1.77 | (0.47, 3.07) | 0.1170 |
| Ferritin (μg/L) | 26.85 | (5.54, 48.16) | 13.85 | (6.71, 20.99) | 0.2433 |
| RBP4 (μmol/L) | 1.63 | (1.44, 1.81) | 1.65 | (1.43, 1.87) | 0.8555 |

*Abbreviations*: AGP, α1-acid glycoprotein; CI, confidence interval; CRP, C-reactive protein; FGF21, fibroblast growth factor 21; I-FABP, intestinal fatty acid–binding protein; IGF-1, insulin-like growth factor 1; IgA, immunoglobulin A; IgG, immunoglobulin G; RBP4, retinol binding protein-4; sCD14, soluble cluster of differentiation 14.

[0] Infants in the non-PRV arm with both anti-rotavirus IgG < 20 units/mL and IgA < 20 units/mL at enrollment, with ≥ threefold rise in IgG and/or IgA after 28 days of follow-up.

[1] Infants in the non-PRV arm with both anti-rotavirus IgG < 20 units/mL and IgA < 20 units/mL at enrollment, with < threefold rise in both IgG and IgA after 28 days of follow-up.

[2] Two-sample Student's t-test with unequal variances.

Table). The only major difference was that in univariable analysis, I-FABP measured by ELISA was significantly associated with a slight increase in HAZ. This association did not maintain statistical significance in the multivariable model.

## MEEDAT biomarkers and natural rotavirus infection in unvaccinated children

There was no evidence of differences in concentrations of I-FABP, sCD14, IGF-1, FGF21, AGP, CRP, ferritin, or RBP4 between children who did and did not experience putative natural rotavirus infection during the 28 days following enrollment (Table 7). All analyses were repeated using I-FABP, sCD14, IGF-1, and FGF21 data from the ELISAs (S7 Table), and no major differences from the original analyses were observed.

## Discussion

Standard ELISAs and the MEEDAT were used to measure I-FABP, sCD14, IGF-1, and FGF21 in sera specimens from Malian children enrolled in a rotavirus vaccine booster trial. Results from the two measurement methods were highly correlated (Table 2), and there was no evidence of concentration-dependent biases in the differences between measurements obtained from the two methods for each biomarker (Fig 3). These performance findings support the viability of analyte quantification in clinical specimens from pediatric populations in low-income countries where EED is prevalent.

Several associations were observed between the selected biomarkers and immune responses to co-administered vaccines and child growth in this Malian cohort. In the quartile analysis, sCD14 levels were negatively associated with the probability of rotavirus seroconversion (based on IgA); however, such an association was not observed in any of the rotavirus RR regression models (Table 3 and Table 5). A negative association between sCD14, a marker of microbial translocation, and protection from oral monovalent rotavirus vaccine (i.e., Rotarix) was previously observed in Bangladeshi infants [12]. No association was observed between

sCD14 concentration and Rotarix seroconversion in a cohort of Zambian infants [31]. Although I-FABP levels in the quartile and univariable RR analyses were negatively associated with the probability of rotavirus seroconversion (based on IgG), the association was not significant in the multivariable RR regression model. Interestingly, the observation of a crude negative association between I-FABP, a marker of enterocyte damage, and PRV seroconversion contrasts sharply with the report of a positive association between I-FABP and Rotarix seroconversion in Zambian infants (31). There are a few plausible explanations for these differences, including possible differences in the immune response to the two rotavirus vaccines as well as the advanced age of the children in the present study compared with the prior studies (9–11 months versus 6–12 weeks). While previous data support the association between markers of EED and poor oral vaccine immunogenicity in young infants [12, 32], the univariable results in the present study suggests that EED may also be associated with poor oral vaccine immunogenicity in older infants (although such associations were not significant after adjustment). In contrast, in older children between 3–14 years of age, a study in Bangladesh observed positive associations between I-FABP and sCD14 with immune responses to oral cholera vaccine [33]. Robust immune correlates of individual protection from rotavirus disease have not been established, however, mean levels of anti–rotavirus IgA in a population were related to the population-level vaccine efficacy against severe disease in prior clinical trials [34, 35]. Although IgG antibodies may be maternally derived and are generally not used in young infants to evaluate immune responses, in this study of older infants, IgG titers may be informative.

In exploratory analyses of parenteral vaccine responses, we observed a significant positive association between IGF-1 concentrations and the probability of YFV seroconversion based on NT in both univariable quartile analysis and univariable RR regression analysis, but not in the multivariable analysis (**Table 4** and **Table 5**). This is the first study to observe crude associations between this potentiator of the GH axis and response to a parenteral vaccine, and is to our knowledge the first study to test the relationship. Although it is possible that the GH axis plays a role in the response to YFV, we hypothesize that IGF-1 in this context serves as a proxy marker of child health and immunocompetence. Accordingly, the significant crude relationship could potentially suggest that children who are producing higher levels of endogenous IGF-1 are more likely to receive protection from parenteral vaccines such as YFV. These results were observed consistently when the analyses were repeated using output from the monoplex ELISAs (**S4 Table**). Although IGF-1 concentrations are highly associated with age, within this cohort that was relatively homogenous with respect to age, IGF-1 and age did not appear to be correlated. However, because IGF-1 and baseline YF titer were negatively correlated (r = -0.1779), adjustment for baseline titer ameliorated the observed univariable association of IGF-1 and YFV seroconversion. A significant negative association was observed between AGP concentrations and the probability of YFV seroconversion in the multivariable RR regression analysis, which suggests that systemic inflammation at the time of immunization reduces the immunogenicity of YFV. However, these exploratory findings must be examined in other settings to confirm their generalizability.

Several biomarkers tested in the study were associated with child growth in regression models. For example, ferritin levels were significantly positively associated with 12-week change in WAZ in univariable analysis but not in multivariable models (**Table 6**), so it is unclear whether this finding is a reflection of confounding from other variables. However this observation contrasts with previous findings in urban Bangladesh of a negative association between ferritin and ponderal growth in infants between 6 and 18 weeks of age [12]. We observed significant negative associations between GLP-2 levels with 12-week change in WAZ and WHZ in both crude and adjusted analyses. Such observations differ from previous reports in rural

Bangladeshi children of positive correlations between GLP-2 levels and 6-month change in WAZ and weight for length z-score between 18 and 24 months of age [36]. One plausible explanation for this difference is that in generally healthy children, GLP-2 could indicate a need for tissue repair (i.e., intestinal damage) and therefore is negatively associated with growth. The malnourished and less healthy children in the Bangladesh study were older and likely needed the hormone to stimulate intestinal tissue repair but those better able (e.g., with more metabolic resources) to produce the hormone grew better. Our observation of significant negative associations between sCD14 levels with 12-week change in WAZ and WHZ in adjusted analyses is consistent with prior reports in a cohort of infants in urban Bangladesh [12], but an association was not observed in a prior investigation in 6–26 month old children in Northeast Brazil [11]. Although previous studies have observed strong negative correlations between AGP, CRP, and sCD14 with IGF-1 in a mixture of stunted and healthy infants [37], sCD14 was positively correlated with IGF-1 in the current study, while AGP and CRP were negatively correlated. Consistent with a previous study in Bangladesh, FGF21 was positively correlated with AGP, suggesting that FGF21 may be in part driven by systemic inflammation [18]. In contrast to that study, FGF21 was not associated with change in child HAZ or WAZ in this Malian cohort; however, the current study was not restricted to malnourished children and did not provide nutritional supplementation. Both I-FABP and sCD14 were positively correlated with AGP in the present study, supporting the hypothesis that intestinal barrier damage and microbial translocation, respectively, contribute to systemic inflammation.

It is unclear whether EED is a risk factor for rotavirus infection in children without immune protection from receipt of the vaccine or natural infection. In our investigation of EED, systemic inflammation, and growth hormone biomarkers, baseline biomarker levels were not significantly different between children who did versus did not have probable rotavirus infection during the study (**Table 7**). However, this analysis was conducted in a relatively small group of children, and therefore warrants further investigation.

This study utilized remnant samples from a well-designed randomized trial and had several notable strengths. The specimens from Mali were collected at enrollment from children in a population at risk for growth deficits and suboptimal vaccine responses, and were recruited from a single setting using a well-defined recruitment protocol and rigorous inclusion and exclusion criteria from the RVI-PRV-01 trial. A major strength in this study is that regular surveillance was conducted on the occurrence of AEs during the first 28 days (e.g., diarrhea, vomiting, etc.). The trial was conducted during rotavirus season in Mali, which raised the possibility that children in the PRV arm could have experienced rotavirus infection after enrollment and prior to the formation of protective immunity from the PRV administered by the study. Therefore, in the rotavirus vaccine seroconversion analyses, children who seroconverted were excluded if it was possible that they had experienced a natural rotavirus infection (i.e., diarrhea, gastroenteritis, or vomiting was reported during the 28 days after vaccination). While stool testing is necessary to confirm rotavirus infection, this combined immunologic and symptomatic approach should reduce the misclassification of seroconversion due to rotavirus infection in the analysis. Rotavirus vaccines (three doses of RotaTeq or two doses of Rotarix) are routinely administered orally in this setting with the first dose at 6 to 12 weeks of age, and subsequent dose(s) administered at 4- to 10-week intervals. For the case-control study, we purposefully selected children without heightened baseline titers (< 20 units/mL) who were therefore less likely to have protection from natural rotavirus infection. In this study, biomarker associations with vaccine(s) seroconversion were tested using both through direct examination of seroconversion by biomarker quartile and then by RR regression using log-2 transformed biomarker concentrations as covariates. Comparing the raw number and proportion of participants who seroconverted in each of the biomarker quartiles is intuitive,

and a nonparametric test for trend in the biomarker quartiles was used to examine associations, however such tests cannot control for potential confounding. In contrast, the RR regression approach allows for non-sample dependent parameterization of the covariates and enables adjustment for the potential confounding effects of the other biomarkers and relevant immunologic measurements (e.g., baseline antibody titers). That is, the multivariable RR regression approach provides an estimate of the association between biomarker levels and seroconversion that is independent of the other covariates. For example, while some of the children in the PRV group already had evidence (heightened baseline IgG and/or IgA titer) of prior infection or immune protection conferred from prior doses of PRV, we accounted for this potential issue by adjusting for baseline anti-rotavirus IgG and anti-rotavirus IgA titer in our multivariable RR regression models. Another strength of this study was the use of a technology suitable for deployment in low-resource setting laboratories to quantitate biomarkers in serum samples with MEEDAT. Use of the Quansys Q-View Imager LS requires a standard wet (e.g., microbiology) laboratory, minimal training of technicians familiar with running ELISA protocols, and a basic laptop computer for data processing and reports. The Imager LS platform is particularly well-suited for use in low-resource settings due to its comparatively low cost, few moving parts, small footprint, and protocol similarity to standard ELISA.

There were a few important limitations to this study. Children in the RVI-PRV-01 were from a narrow age range and were relatively healthy, somewhat limiting the generalizability of our conclusions. Therefore, results must be interpreted cautiously and confirmed in other populations. Several IGF-1 values fell more than 20% below the LLOQs for both MEEDAT and ELISA, as did several I-FABP values on ELISA. Owing to resource and sample volume limitations, it was not possible to repeat IGF-1 ELISAs, I-FABP ELISAs, or MEEDAT (for IGF-1 only) without dilution where results were out of quantification range. Although the performance testing results (i.e., Pearson's correlation and Bland-Altman plots) for the MEEDAT and ELISA included nearly all samples for sCD14 and FGF21, such results for IGF-1 and I-FABP were restricted to the 235 and 230 samples, respectively, that were in range of the ELISA (IGF-1 > 7.5 ng/mL; I-FABP > 376 pg/mL) and MEEDAT (IGF-1 > 2.7 ng/mL). With the exception of sCD14, the biomarker concentrations exhibited a heavily skewed distribution, with relatively few high values to contribute to the performance assessment. Therefore, additional testing on specimens with higher biomarker concentrations may benefit our understanding of the MEEDAT's comparative performance. Resource limitations restricted the measurement of GLP-2 to only a subset of the 300 children, but these specimens were intentionally selected from children with low baseline levels (< 20 units/mL) of anti-rotavirus IgA and IgG in order to maximize their relevance for the rotavirus seroconversion analysis. Exclusions due to probable rotavirus infection were done after GLP-2 testing, thus reducing the sample size from 152 to 133. While this sample size could have been larger, because such exclusions were not made before testing, the analyses of GLP-2 with YFV and MenAV seroconversion were less subject to bias. The significant associations between IGF-1 with YFV seroconversion and between I-FABP with rotavirus IgG seroconversion were only observed in unadjusted analyses; however, such findings warrant further investigation. Although numerous statistical tests are run and results presented in the study (e.g., Tables 3–6), no formal effort was made to control for the family-wise error rate such as the use of the Benjamini–Hochberg or Bonferroni method for alpha-correction. However, the use of unadjusted P-values is acceptable in exploratory analyses such as those in this study. Case-control sample sizes were driven by sample availability from non-PRV recipients who had a rise in anti-rotavirus IgA or IgG over 28 days, and this sample size may not have provided adequate power to detect a statistically significant difference in baseline biomarker concentrations. Future work to answer this important question will need to come from a larger (preferably prospective) cohort of non-

immunized children. Potentially useful data on gestational age, birth weight, and birth length were not possible to obtain because all children were recruited and enrolled at a minimum age of 9 months, and data were not collected on maternal antibodies (systemic or in breast milk) against rotavirus, MenA, and YF.

In conclusion, MEEDAT performed well in comparison to standard ELISAs for the measurement of four analytes for EED and GH resistance. MEEDAT may be a rapid and effective tool for efficiently screening children for EED prior to enrollment into clinical trials of candidate EED interventions. Furthermore, information from MEEDAT could help streamline the evaluation of efficacy in trials in which the biomarkers of growth hormone axis, systemic inflammation, and EED are clinical endpoints. In the next several years, MEEDAT has the potential to reduce the time and cost and increase accuracy of serum/plasma-based biomarker measurement among children at risk for/suffering from malnutrition in clinical research. Adoption of MEEDAT could help accelerate the identification of interventions that prevent or treat child stunting and/or boost the immunogenicity of vaccinations given to children in low-resource settings.

## Supporting information

**S1 Table. Descriptive information, rationale, and previously observed pediatric concentrations for the selected EED or GH axis biomarkers.**
(DOCX)

**S2 Table. Pairwise correlations between biomarkers of environmental enteric dysfunction, the growth hormone axis, systemic inflammation, and micronutrient status.**
(DOCX)

**S3 Table. Rotavirus vaccine seroconversion at 28 days post-immunization by EED and GH status measured by ELISA, in infants without natural rotavirus infection during follow-up (n = 220 unless otherwise noted).**
(DOCX)

**S4 Table. Yellow fever and meningococcal A vaccine seroconversion at 28 days post-immunization by EED and GH status measured by ELISA (N = 300 unless otherwise noted).**
(DOCX)

**S5 Table. Associations between baseline log-2 transformed serum biomarker concentrations (ELISA) with 28-day seroconversion to rotavirus, yellow fever, and meningococcal A vaccines.**
(DOCX)

**S6 Table. Associations between baseline log-2 transformed serum biomarker concentrations (ELISA) with change in HAZ, WAZ, and WHZ over 12 weeks of follow up.**
(DOCX)

**S7 Table. Comparison of baseline serum biomarker concentrations (by ELISA) in children with vs. without putative rotavirus infection in 28 days of follow up.**
(DOCX)

**S1 Data. All clinical and laboratory data used in manuscript analyses.**
(CSV)

**S2 Data. Dictionary to accompany the clinical and laboratory data in S8 Data.**
(XLSX)

## Acknowledgments

We thank the RVI-PRV-01 participants and their families as well as clinical and regulatory staff at CVD-Mali (Mamadou Diallo and Rokiatou Dembele) and the University of Maryland (Milagritos Tapia, Samba Sow, Kathleen Neuzil, and Karen Kotloff), for the generous contribution of their time and effort. We are grateful for the time, effort, and guidance provided by PATH clinical staff, most notably Corey Kelly, Niranjan Bhat, and Yuxiao Tang. We are grateful for the considerable time and effort contributed by the respective laboratories of Monica McNeal at Cincinnati Children's Hospital, Cristina Domingo Carrasco at the Robert Koch Institute, and Ray Borrow at Public Health England. Special thanks to our collaborators at Quansys Biosciences for their research and development work, especially Abby Tyler, Chris Lyman, and Meghan Young. We also thank Donna Denno and Judd Walson for their advice and support in this work, and William Petri Jr., Kerry Schulze, Andrew Prendergast, Jennie Ma, Uma Nayak, Richard Guerrant, and Relana Pinkerton for providing pediatric biomarker data used to specify the clinical range for the EED and GH axis analytes.

## Author Contributions

**Conceptualization:** Michael B. Arndt, Michael Kalnoky, David S. Boyle, Eugenio L. de Hostos, Robert K. M. Choy.

**Data curation:** Michael B. Arndt, Heather N. White, Gregory Bizilj.

**Formal analysis:** Michael B. Arndt, Laina D. Mercer, Heather N. White, Gregory Bizilj.

**Funding acquisition:** Eugenio L. de Hostos, Robert K. M. Choy.

**Investigation:** Michael B. Arndt, Jason L. Cantera, Heather N. White, Gregory Bizilj, Robert K. M. Choy.

**Methodology:** Michael B. Arndt, Jason L. Cantera, Laina D. Mercer, Michael Kalnoky.

**Project administration:** David S. Boyle, Eugenio L. de Hostos.

**Software:** Michael B. Arndt, Heather N. White.

**Supervision:** Laina D. Mercer, Robert K. M. Choy.

**Writing – original draft:** Michael B. Arndt.

**Writing – review & editing:** Michael B. Arndt, Jason L. Cantera, Laina D. Mercer, David S. Boyle, Eugenio L. de Hostos, Robert K. M. Choy.

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
