## [Decision Letter · Decision Letter 0]

26 Nov 2019

Dear Dr. Arndt:

Thank you very much for submitting your manuscript "Validation of the Micronutrient and Environmental Enteric Dysfunction Assessment Tool and evaluation of biomarker risk factors for growth faltering and vaccine failure in young Malian children" (#PNTD-D-19-01273) for review by PLOS Neglected Tropical Diseases. Your manuscript was fully evaluated at the editorial level and by independent peer reviewers. The reviewers appreciated the attention to an important problem, but raised some substantial concerns about the manuscript as it currently stands. These issues must be addressed before we would be willing to consider a revised version of your study. We cannot, of course, promise publication at that time.

We therefore ask you to modify the manuscript according to the review recommendations before we can consider your manuscript for acceptance. Your revisions should address the specific points made by each reviewer. In addition, please make sure that you add a supplemental data file to the submission that will allow others to reproduce your analyses in line with the PLOS Data Policy. 

When you are ready to resubmit, please be prepared to upload the following:

(1) A letter containing a detailed list of your responses to the review comments and a description of the changes you have made in the manuscript.

(2) Two versions of the manuscript: one with either highlights or tracked changes denoting where the text has been changed (uploaded as a "Revised Article with Changes Highlighted" file); the other a clean version (uploaded as the article file).

(3) If available, a striking still image (a new image if one is available or an existing one from within your manuscript). If your manuscript is accepted for publication, this image may be featured on our website. Images should ideally be high resolution, eye-catching, single panel images; where one is available, please use 'add file' at the time of resubmission and select 'striking image' as the file type. 

Please provide a short caption, including credits, uploaded as a separate "Other" file. If your image is from someone other than yourself, please ensure that the artist has read and agreed to the terms and conditions of the Creative Commons Attribution License at http://journals.plos.org/plosntds/s/content-license (NOTE: we cannot publish copyrighted images). 

(4) If applicable, we encourage you to add a list of accession numbers/ID numbers for genes and proteins mentioned in the text (these should be listed as a paragraph at the end of the manuscript). You can supply accession numbers for any database, so long as the database is publicly accessible and stable. Examples include LocusLink and SwissProt.

(5) To enhance the reproducibility of your results, we recommend that you deposit your laboratory protocols in protocols.io, where a protocol can be assigned its own identifier (DOI) such that it can be cited independently in the future. For instructions see http://journals.plos.org/plosntds/s/submission-guidelines#loc-methods

While revising your submission, please upload your figure files to the Preflight Analysis and Conversion Engine (PACE) digital diagnostic tool, https://pacev2.apexcovantage.com/ PACE helps ensure that figures meet PLOS requirements. To use PACE, you must first register as a user. Then, login and navigate to the UPLOAD tab, where you will find detailed instructions on how to use the tool. If you encounter any issues or have any questions when using PACE, please email us at figures@plos.org.

We hope to receive your revised manuscript by Jan 25 2020 11:59PM. If you anticipate any delay in its return, we ask that you let us know the expected resubmission date by replying to this email.

To submit a revision, go to https://www.editorialmanager.com/pntd/ and log in as an Author. You will see a menu item call Submission Needing Revision. You will find your submission record there. 

Sincerely,

Andrew S. Azman

Deputy Editor

Reviewer's Responses to Questions

**Key Review Criteria Required for Acceptance?**

**Methods**

-Are the objectives of the study clearly articulated with a clear testable hypothesis stated?

-Is the study design appropriate to address the stated objectives?

-Is the population clearly described and appropriate for the hypothesis being tested?

-Is the sample size sufficient to ensure adequate power to address the hypothesis being tested?

-Were correct statistical analysis used to support conclusions?

-Are there concerns about ethical or regulatory requirements being met?

Reviewer #1: The manuscript focuses on associations between four different outcomes and multiple biomarkers, each measured in two different ways (MEEDAT versus ELISA) and using two different methods of statistical analysis. It is therefore very hard to take in all these findings. Furthermore, the findings are generally over-interpreted with too much emphasis on P values rather than effect sizes with an accompanying measure of precision, and equal weight given in interpretation to univariable and multivariable analysis. There is a general lack of hypotheses throughout; for example, it is unclear why the authors would evaluate associations between IGF-1 and yellow fever vaccine seroconversion: what is the hypothesis underlying this question?

Reviewer #2: 1. Are the objectives of the study clearly articulated with a clear testable hypothesis stated?

Adding a line clearly defined the objectives of this work is missing and should ideally be placed at the end of introduction

2. -Is the study design appropriate to address the stated objectives?

Flow chart for study design is missing

3. Is the population clearly described and appropriate for the hypothesis being tested?

yes, but a supplementary table would help clearly identifying group with numbers in each sub analyses group, they have used different numbers for sub analyses

4. s the sample size sufficient to ensure adequate power to address the hypothesis being tested?

 yes adequate, though sample size calculation is not given

5. -Were correct statistical analysis used to support conclusions?

yes- suggest to get it reviewed by a Biostatician- I cannot make comments on Regression on Log2 transformation, if it is appropriate?

6. Are there concerns about ethical or regulatory requirements being met?

all ethical approval are taken. in case of bio-repository specimen, was ethical exemption obtained?

Reviewer #3: The Methods appeared adequate.

**Results**

-Does the analysis presented match the analysis plan?

-Are the results clearly and completely presented?

-Are the figures (Tables, Images) of sufficient quality for clarity?

Reviewer #1: See above

Reviewer #2: Does the analysis presented match the analysis plan?

it is adequate but a reviewer or statistician should review it for its correct interpretation. This is beyond my expertise. however, authors has not mentioned if p values are FDR corrected? 

-Are the results clearly and completely presented?

Results are presented clearly. I would suggest to add a figure of Meedat platform, a pictorial will be helpful for a regular panel and analytes that were used in this study. I donot understand mentioning of micro-nutrient panel in abstract as these results were not shown in the manuscript at all. Please clarify if these analytes were tested on these samples and results of only 4 GH resistance markers are described in this paper? it seems that there are 18 analytes per well.

-Are the figures (Tables, Images) of sufficient quality for clarity?

titles of tables are quite long, also figure legend is missing for figure 1 and 2. 

table S3 and S4 are ELISA based results but title of table 4 and 5 are very much similar to suppl tables. please specify if table 4 and 5 are only MEEDAT results ? why suppl. tables have less biomarkers tested by ELISA?

it would be better to mention table # in results and in discussion section. Reader seem to get easily lost as what analyses author is referring to?

Abstract: line 34, WHZ instead of weight for height

line 19: micronutrient deficiency can be removed 

line124, pg 7: 1 biomarker can be changed to single biomarker

Method: line 174, please give "n" for children excluded due to natural rota virus infection

line 230 and 231, pg 11: mention the name of labs where NT and SBA were performed.

line 306, page 15: please add "using a Poisson working .."

line 307, pg 15: "are" 

table 3: Only female (47.7%) are give in this table. Is it assumed that 52.3% are male. Why male was not mentioned here? 

line 543, pg 29: WFH, is it WHZ?

I am wondering whether MEEDAT and ELISA median levels were compared by MWU test? please add column for p value. Anthropometric comparison for enrollment and 12 weeks can be done parallel to each other characteristics should not be repeated

Reviewer #3: The Results appeared adequate.

**Conclusions**

-Are the conclusions supported by the data presented?

-Are the limitations of analysis clearly described?

-Do the authors discuss how these data can be helpful to advance our understanding of the topic under study?

-Is public health relevance addressed?

Reviewer #1: See above

Reviewer #2: -Are the conclusions supported by the data presented?

yes field applicability of MEEDAT should be discussed

-Are the limitations of analysis clearly described?

yes adequately described

-Do the authors discuss how these data can be helpful to advance our understanding of the topic under study?

yes, adequate discussion in conclusion

-Is public health relevance addressed?

Application of technology in terms of filed application needs to be discussed such as a standard microbiology lab equipped with standard equipment can perform and interpret this test. Does this test require esp equipment, esp operational needs, training os staff etc?. 

Discussion is too long. please summarize important finding with support of data. I suggest a way to present comparison of MEEDAT data in the first part and serconversion in the second part.

Reviewer #3: The Conclusions appeared adequate.

**Editorial and Data Presentation Modifications?**

Reviewer #1: The paper needs a Figure that explains the different cohorts, substudies and numbers.

Fig 1 – the dotted lines for the 2SD threshold are not coming out clearly, they are indistinguishable from the mean difference line

Reviewer #2: Abstract: line 34, WHZ instead of weight for height

line 19: micronutrient deficiency can be removed 

line 124, pg 7: 1 biomarker can be changed to single biomarker

Method: line 174, please give "n" for children excluded due to natural rota virus infection

line 230 and 231, pg 11: mention the name of labs where NT and SBA were performed.

line 306, page 15: please add "using a Poisson working .."

line 307, pg 15: "are" 

table 3: I am wondering whether MEEDAT and ELISA median levels were compared by MWU test? please add column for p value. Anthropometric comparison for enrollment and 12 weeks can be done parallel to each other characteristics should not be repeated

line 543, pg 29: WFH, is it WHZ

line 601, pg 33: selected biomarkers

Reviewer #3: Dr. Arndt et al. present a report detailing multiple biomarkers of environmental enteric dysfunction (EED) and growth in a cohort of Malian children assessed for vaccine response. They make comparisons between the multiplex assay and more standard ELISA assays and compare to growth and vaccine response outcomes. This is an important topic relevant to the assessment of children in developing areas of the world.

I do have several concens:

1. The authors in Tables 4 and 5 assess the relationship between seven different markers of EED (as quartiles of each marker) and seroconversion prevalence for Rotavirus IgA and IgG (Table 4) and seroconversion to Yellow Fever Virus or meningococcal A virus (Table 5), representing a total of 28 assessments. There are then an additional 56 analyses in Table 6 and 48 analyses in Table 7. There were p-values <0.05 for one of each of Rotavirus IgA, IgG and Yellow Fever Virus seroconversion in Tables 4 and 5 and for three additional analyses in Table 6 and 7 in Table 7. A few of these associations did exhibit very low p values. Nevertheless, several other associations had higher p-values closer to 0.05, as a reminder that the potential still remained that these associations could have happened by chance. With the number of assessments being performed, the authors should mention in their limitations section the potential for some spurious associations found because of multiple comparisons. 

2. In the Discussion (lines 627-637), the authors note the positive association between levels of IGF-1 and immune response to Yellow Fever Virus that was in the univariable model but not in the multivariable model. Among other factors, levels of IGF-1 are highly associated with advancing age, and the adjustment for age could have resulted in the amelioration of the univariable association. This possibility should be mentioned in the text.

Minor:

Line 481: In discussing the probability of rotavirus seroconversion based on sCD14 quartile, it appears that the p values for IgA and IgG have been reversed in the text.

**Summary and General Comments**

Reviewer #1: The manuscript by Arndt and colleagues focuses on environmental enteric dysfunction (EED), which is a highly prevalent but ill-understood disorder of low- and middle-income countries. Overall, they present an interesting set of analyses, focusing particularly on a new tool (MEEDAT) which evaluates a range of putative EED biomarkers in 10uL serum. This paper combines an evaluation of the MEEDAT performance with four distinct questions regarding EED and child health, growth and vaccine immunogenicity. As such, the manuscript is complex, too long, and hard to navigate since it loses focus with so many substudies. There is a need to substantially overhaul the manuscript to focus attention on one or two of these areas, because it is difficult for readers to take in all the information provided as it is currently presented.

The paper needs to be substantially simplified, since the cohorts, biomarkers and questions are of great relevance and interest, and the analyses generally sound, but it is currently un-navigable as a reader. I would suggest focusing on fewer questions, and limiting the interpretation to consistent findings that remain significant in multivariable analyses, then synthesizing the findings better for the reader.

Minor comments:

1. Introduction lines 66-67 – update estimates based on 2018 data

2. A photograph of the MEEDAT system would be very helpful for readers unfamiliar with the Q Plex system

3. Since this paper compares MEEDAT to ELISA, it is important to present quality control data on how well the ELISAs performed (e.g. intra- and inter-plate CVs for each assay)

4. Methods line 292 – what about implausible gains in height? If these were not also dropped from the dataset then it becomes biased 

5. Table 8 – the comparison of children with (N=27) and without (N=27) rotavirus infection is underpowered and relies on a putative measure of rotavirus infection. The paper is stronger without this substudy.

Reviewer #2: It is a well written manuscript on a very important topic of public health research. The presentation of data can be improved by adding flow chart about study subjects. I struggled a lot to compare numbers and missing data, for eg. suppl. tables S3 and S4 donot match with number of biomarkers shown in main tables 4 & 5. Both method section and discussion are way too long, both needs to be concise and focused. In my view the presentation of results can be improved and words count can be decreased.

1- A general introduction of MEEDAT platform in a pictorial or a graphical abstract will help reader to understand it. I believe there are lot of anlaytes that can be tested but author can identify which analytes they are focusing for the current manuscript. 

2- It is important to add few sentences about field applicability of this assay. what type of lab infrastructure is needed to perform this test to assess the user friendly aspect of this test. 

3- It seems that there are two or three major objectives of this study. It would be nice if the objectives are clearly written in introduction section and results are aligned with those objectives. 

4- Regarding affect of EED biomarkers on YFV and MV, please give the rationale in introduction as they are not oral vaccine. 

5-Baseline samples were shipped to Seattle for analyses. Rationale of collecting only baseline samples for EED study is not cleared to me, please mention if EED biomarkers were again measured at 28 day?

6-please specify if the biomarker data was previously published esp ELISA results?

7- Please discuss the difference in ages of bio repository and Mali study. What are the implications for pooling these results, if bio-markers are age specific?

Reviewer #3: (No Response)

PLOS authors have the option to publish the peer review history of their article (what does this mean?). If published, this will include your full peer review and any attached files.

Reviewer #1: No

Reviewer #2: Yes: Najeeha Talat Iqbal

Reviewer #3: No

---

## [Decision Letter · Decision Letter 1]

3 Jun 2020

Dear Dr. Arndt,

Thank you very much for submitting your manuscript "Validation of the Micronutrient and Environmental Enteric Dysfunction Assessment Tool and evaluation of biomarker risk factors for growth faltering and vaccine failure in young Malian children" for consideration at PLOS Neglected Tropical Diseases. As with all papers reviewed by the journal, your manuscript was reviewed by members of the editorial board and by several independent reviewers. The reviewers appreciated the attention to an important topic. Based on the reviews, we are likely to accept this manuscript for publication, providing that you modify the manuscript according to the review recommendations. 

Sincerely,

Shaden Kamhawi

Editor-in-Chief 

The editors would like the authors to address the comments of reviewers 1 and 3 by including a few sentences highlighting the limitation of their study.

Reviewer's Responses to Questions

**Key Review Criteria Required for Acceptance?**

**Methods**

-Are the objectives of the study clearly articulated with a clear testable hypothesis stated?

-Is the study design appropriate to address the stated objectives?

-Is the population clearly described and appropriate for the hypothesis being tested?

-Is the sample size sufficient to ensure adequate power to address the hypothesis being tested?

-Were correct statistical analysis used to support conclusions?

-Are there concerns about ethical or regulatory requirements being met?

Reviewer #1: See below

Reviewer #2: -Are the objectives of the study clearly articulated with a clear testable hypothesis stated?

it has now been added in intro section

-Is the study design appropriate to address the stated objectives?

yes

-Is the population clearly described and appropriate for the hypothesis being tested?

yes

-Is the sample size sufficient to ensure adequate power to address the hypothesis being tested?

still under power but for exploratory study may be considered

-Were correct statistical analysis used to support conclusions?

yes

-Are there concerns about ethical or regulatory requirements being met?

no

Reviewer #3: (No Response)

**Results**

-Does the analysis presented match the analysis plan?

-Are the results clearly and completely presented?

-Are the figures (Tables, Images) of sufficient quality for clarity?

Reviewer #1: See below

Reviewer #2: -Does the analysis presented match the analysis plan?

yes

-Are the results clearly and completely presented?

yes

-Are the figures (Tables, Images) of sufficient quality for clarity?

yes

Reviewer #3: (No Response)

**Conclusions**

-Are the conclusions supported by the data presented?

-Are the limitations of analysis clearly described?

-Do the authors discuss how these data can be helpful to advance our understanding of the topic under study?

-Is public health relevance addressed?

Reviewer #1: See below

Reviewer #2: -Are the conclusions supported by the data presented?

yes

-Are the limitations of analysis clearly described?

yes

-Do the authors discuss how these data can be helpful to advance our understanding of the topic under study?

yes it has now been added

-Is public health relevance addressed?

yes it has now been added

Reviewer #3: (No Response)

**Editorial and Data Presentation Modifications?**

Reviewer #1: See below

Reviewer #2: none

Reviewer #3: (No Response)

**Summary and General Comments**

Reviewer #1: The authors have addressed most of my original comments, included a flow chart which helps the reader to follow the different participants, and removed data from the US cohort. This generally helps to simplfy the manuscript.

I think the authors still place too much emphasis on their univariable results in their interpretation. For example, lines 575-576 of the discussion do not reflect the data shown in Table 5, where none of the adjusted RR of rotavirus vaccine seroconversion suggest an association with EED, whereas the conclusion states that EED may impair rotavirus vaccine responses in this older infant cohort. Similarly, lines 584 onwards still consider the IGF-1 findings to be important despite the fact that this association is no longer present in an adjusted model. I think this is an over-optimistic interpretation of the data, which largely shows no relationship between EED and vaccine responses. 

There are two minor mistakes I think:

1. Line 467 seems to have an error in the 95%CI

2. Line 33 of abstract - this is not a positive association. Higher AGP is associated with a lower relative risk of seroconversion

Reviewer #2: all responses are adequately answered

Reviewer #3: The authors adequately addressed my concerns in their revision.

PLOS authors have the option to publish the peer review history of their article (what does this mean?). If published, this will include your full peer review and any attached files.

Reviewer #1: No

Reviewer #2: No

Reviewer #3: No
---

## [Editor Report · Decision Letter 2]

24 Jul 2020

Dear Dr. Arndt,

Thank you very much for re-submitting your manuscript "Validation of the Micronutrient and Environmental Enteric Dysfunction Assessment Tool and evaluation of biomarker risk factors for growth faltering and vaccine failure in young Malian children" for consideration at PLOS Neglected Tropical Diseases. We are close to accepting your manuscript for publication but would like you to please resubmit a revised version either attaching the data needed to reproduce the analyses as a supplement, or by providing a url to a publicly available repository. 

Sincerely,

Andrew S. Azman

Deputy Editor
---

## [Editor Report · Decision Letter 3]

13 Aug 2020

Dear Dr. Arndt,

We are pleased to inform you that your manuscript 'Validation of the Micronutrient and Environmental Enteric Dysfunction Assessment Tool and evaluation of biomarker risk factors for growth faltering and vaccine failure in young Malian children' has been provisionally accepted for publication in PLOS Neglected Tropical Diseases.

Best regards,

Andrew S. Azman

Deputy Editor

Andrew Azman

Deputy Editor

---

## [Editor Report · Acceptance letter]

24 Sep 2020

Dear Dr. Arndt,

We are delighted to inform you that your manuscript, "Validation of the Micronutrient and Environmental Enteric Dysfunction Assessment Tool and evaluation of biomarker risk factors for growth faltering and vaccine failure in young Malian children," has been formally accepted for publication in PLOS Neglected Tropical Diseases.

Best regards,

Shaden Kamhawi

co-Editor-in-Chief

Paul Brindley

co-Editor-in-Chief
